# Layered feedback control overcomes performance trade-off in synthetic biomolecular networks

Chelsea Y. Hu [1,2] ✉ & Richard M. Murray [1]

Layered feedback is an optimization strategy in feedback control designs widely used in engineering. Control theory suggests that layering multiple feedbacks could overcome the robustness-speed performance trade-off limit. In natural biological networks, genes are often regulated in layers to adapt to environmental perturbations. It is hypothesized layering architecture could also overcome the robustness-speed performance trade-off in genetic networks. In this work, we validate this hypothesis with a synthetic biomolecular network in living E. coli cells. We start with system dynamics analysis using models of various complexities to guide the design of a layered control architecture in living cells. Experimentally, we interrogate system dynamics under three groups of perturbations. We consistently observe that the layered control improves system performance in the robustness-speed domain. This work confirms that layered control could be adopted in synthetic biomolecular networks for performance optimization. It also provides insights into understanding genetic feedback control architectures in nature.

Two of the major goals of synthetic biology are to engineer biological systems to perform desired tasks and to understand biomolecular networks by building them. These two goals are both very ambitious; fortunately, their advancements are strongly coupled. By building and studying synthetic systems, we gain insights into the natural networks; with a deeper understanding of how natural genetic networks allow life to persevere through constant adversity, we develop stronger theoretical principles to guide synthetic network design. Like many other systems humans have engineered, feedback controls are essential components of a reliable synthetic biological system. Over the past two decades, the synthetic biology toolbox of characterized genetic components, modules, and motifs has been expanding exponentially[1]. As the complexity of synthetic circuits grows, its interplay with control theory becomes more prominent. A significant amount of work has already shown that applying feedback controls improves the reliability, efficiency, and performances of synthetic biological systems[2–8]. Feedback control provides correcting actions based on the difference between desired and actual performance[9,10], therefore it buffers systems from external disturbances and variations of components within

the system. However, feedback can also destabilize the system when improperly designed[10]. Feedback design is especially challenging in biological systems due to its complexity – all molecular species are part of an extensive endogenous network that consists of numerous feedback mechanisms.

In nature, biology has developed remarkably sophisticated strategies to apply feedback controls to biomolecular and physiological networks. Interestingly, layering and redundancy appears to be a common style in these feedback architectures. For instance, bacteria cells layer a positive control to move towards nutrients and a negative control to move away from toxins through chemotaxis signaling[11,12]. During a heat shock, the endogenous control in bacteria maintains the amount of heat shock proteins with a multi-layer control strategy via translation, chaperone interaction, and protein degradation[13]. At the physiological level, the human body maintains a relatively constant glucose level in the bloodstream through insulin, the production of which is regulated through the interplay of the pancreas with the brain, liver, gut, as well as adipose and muscle tissues[14]. It has also been discovered that the sleep and arousal states of animals are controlled

[1]Division of Biology and Biological Engineering, California Institute of Technology, Pasadena, CA, USA. [2]Department of Chemical Engineering, Texas A&M University, College Station, TX, USA. ✉e-mail: chelsea.hu@tamu.edu

with a layered architecture[15]. Control theorists found that layering is a powerful optimization method for feedback control design[16]. This is because the performance of feedback controls is often bound by hard limits. A system that is optimized for one type of disturbance is typically fragile to other types of disturbance[10,17]. These constraints could be manifested as the robustness-efficiency trade-off[18], the speed-accuracy trade-off[19–21], and the speed-robustness trade-off[20]. Previous theoretical work also suggests that in both natural and engineered systems, if multiple control modules of various performance profiles are layered together, this hard limit could be overcome[19,21,22].

In this work, we set to investigate the robustness-speed performance trade-off of biomolecular feedback control and the effects of layered architecture. Here we define speed by the inverse of settling time and robustness by the inverse of peak disturbance of a system when subjected to an impulse or step perturbation. To simplify the problem, we forward engineer layered feedback control in living E. coli to study the dynamical performance of this architecture in a biomolecular context. We first start with a node-based design of two layers of feedback (Fig. 1b) and a minimal model of three species (Eqs. (1)–(3)). By performing system analysis on the linearized state space model, we observe the robustness-speed trade-off on the two single-layer designs, where one appears to be robust but slow while the other is fast but fragile. Meanwhile, we see a clear integration of these two traits in the layered feedback system. Subsequently, we expand this design into a generic biomolecular model that describes the coarse-grained dynamics of a biomolecular feedback system. With simulation, we also observe that for all four sets of the proposed designs, the two single-layer feedbacks are bound by the robustness-speed trade-off limit, while the layered feedback design overcomes it. Based on the simulated result, we choose the best performing layered feedback design and construct it in E. coli. Finally, we present six sets of dynamical perturbation experiments, with chemical, temperature, and nutrient disturbances, each on two directions. We show the layered control design improves system performances by integrating the response profiles of the single-layer controllers. This result applies layered feedback control in a biomolecular network and validates a key hypothesis of layered control theory. In addition to providing a validated optimization strategy to genetic network engineering, we hope the insights it provides will improve our understanding of natural dynamical systems in biology.

## Results

### Robustness-speed trade-off analysis with a node-based design

First, to simplify the problem, we proposed a layered negative feedback control architecture with a node-based design for system analysis. As shown in Fig. 1a, we defined two nodes, A and B, where B is the observable output of the system that is activated by A. Species R is a byproduct of species B that negatively regulates the expression of B and itself through two possible routes: cis feedback (R represses B) and trans feedback (R represses A). The activation and repression here are estimated with first-order Hill functions:

$$\frac{dA}{dt} = \gamma \cdot \beta_A \cdot f_{trans} - d \cdot A \tag{1}$$

$$\frac{dR}{dt} = \gamma \cdot \beta_R \cdot \left(\frac{A}{K_A + A}\right) \cdot f_{cis} - d_R \cdot R \tag{2}$$

$$\frac{dB}{dt} = \gamma \cdot \beta_B \cdot \left(\frac{A}{K_A + A}\right) \cdot f_{cis} - d \cdot B \tag{3}$$

For the open loop, $f_{cis} = f_{trans} = 1$; for the cis only feedback, $f_{cis} = \frac{K_R}{K_R + R}$, $f_{trans} = 1$; for the trans only feedback, $f_{trans} = \frac{K_R}{K_R + R}$, $f_{cis} = 1$; for the layered feedback, $f_{cis} = f_{trans} = \frac{K_R}{K_R + R}$. Both A and B were assumed to be

protein species, while R could be either protein or regulatory RNA. In the ODEs, $\beta_A$, $\beta_R$, and $\beta_B$ denote the expression rate of A, R, and B, respectively. Here, we set $\beta_A = 1$, $\beta_B = \beta_R = 5$. The constant $K_A$ defines the activation coefficient of A, and $K_R$ defines the repression coefficient of R. Both species A and B degrade at the same rate $d$, while R degrades at rate $d_R$. We defined that if R is a protein species, then $d_R = d$, if R is an RNA species, then $d_R = 10 \cdot d$ for faster degradation. To ensure that the protein and the RNA regulator have compatible regulating effectiveness, we constrained the product of $K_R \cdot d_R$ to be constant (i.e. for an RNA regulator $K_R = 10$, $d_R = 0.3$, and for a protein regulator $K_R = 100$, $d_R = 0.03$). Here we defined the output as species B. The input was defined as $\gamma$, a unitless scalar that impacts the expression rate of all three species ($\beta_A$, $\beta_R$, and $\beta_B$). Input $\gamma$ is also the disturbance source of interest and was initially set to 1. Our analysis in the rest of this work focuses on four designs: open loop, the cis feedback, the trans feedback, and the layered feedback.

In Fig. 1c, we obtained the system's Bode plot gain curve to describe the design's system dynamics. A Bode plot is a common tool used in control theory to visualize a linear system's dynamics in respect to the input's frequency[10]. Since all four systems are nonlinear, we linearized them at their equilibrium points before converting them to the frequency domain (see Methods). In this plot, the x-axis represents the input frequency in rad/min, and the y-axis represents the output/input magnitude, which has a unit of nM. When the input frequency is low, this could be understood as a long-lasting impulse perturbation to the universal species production scalar $\gamma$. When the input frequency is high, the input signal oscillates rapidly, and the output response to the disturbances diminishes. In this figure, we observed that all three types of feedback attenuate disturbances at low input frequency. In which, the cis feedback (red) has a stronger attenuation effect than the trans feedback (yellow), and the layered feedback shows the strongest attenuation effect (purple). As the input frequency increases, the trans feedback loses its disturbance attenuation property compared to the open loop. However, the cis feedback control remains more robust than the open loop in the entire frequency span. We also observed that there is only a slight advantage for a protein regulator (dashed lines) relative to an RNA regulator (solid lines) at intermediate input frequencies. Additionally, the output magnitudes of the two proposed architectures (R as an RNA regulator and R as a protein regulator) when responding to low frequency disturbances are indistinguishable.

To investigate the impact of species R on the system dynamics, we analyzed the peak disturbance and settling time of each system under both step and impulse perturbations, with a 10 × 10 parameter space that defines the property of regulator R. In Fig. 1d, e, we showed the step response analysis of the four systems. A large $K_R$ (y-axis) models a weak regulator, and a large $d_R$ (x-axis) models a fast-degrading regulator. RNA species usually degrade faster than protein species. We found that over a wide parameter space, the cis feedback attenuates disturbances at low frequency better than the trans feedback, while the layered feedback shows the most attenuation. In each feedback design, regulators with strong repressive strength and a slow degradation rate achieve the most disturbance attenuation (Fig. 1d upper panel). In the lower panel of Fig. 1d, we plotted the settling time of each construct in the same two-dimensional parameter space $K_R$ (y-axis) and $d_R$ (x-axis). The system performance in this parameter scan is also demonstrated in the robustness and speed dimensions, as shown in Fig. 1e. The combination of Fig. 1d, e showed that, under a step perturbation, the cis feedback shows a strong attenuation of disturbance yet takes longer to settle to equilibrium; the trans feedback settles faster but attenuates the disturbances less effectively. To study the systems' performance with a high-frequency perturbation, we also analyzed the impulse response of these four designs in the same parameter space, as shown in Supplementary Fig. 3 and Fig. 1f. The analysis showed a similar pattern in the speed-robustness dimension, with the trans feedback demonstrating no attenuation effect, which is

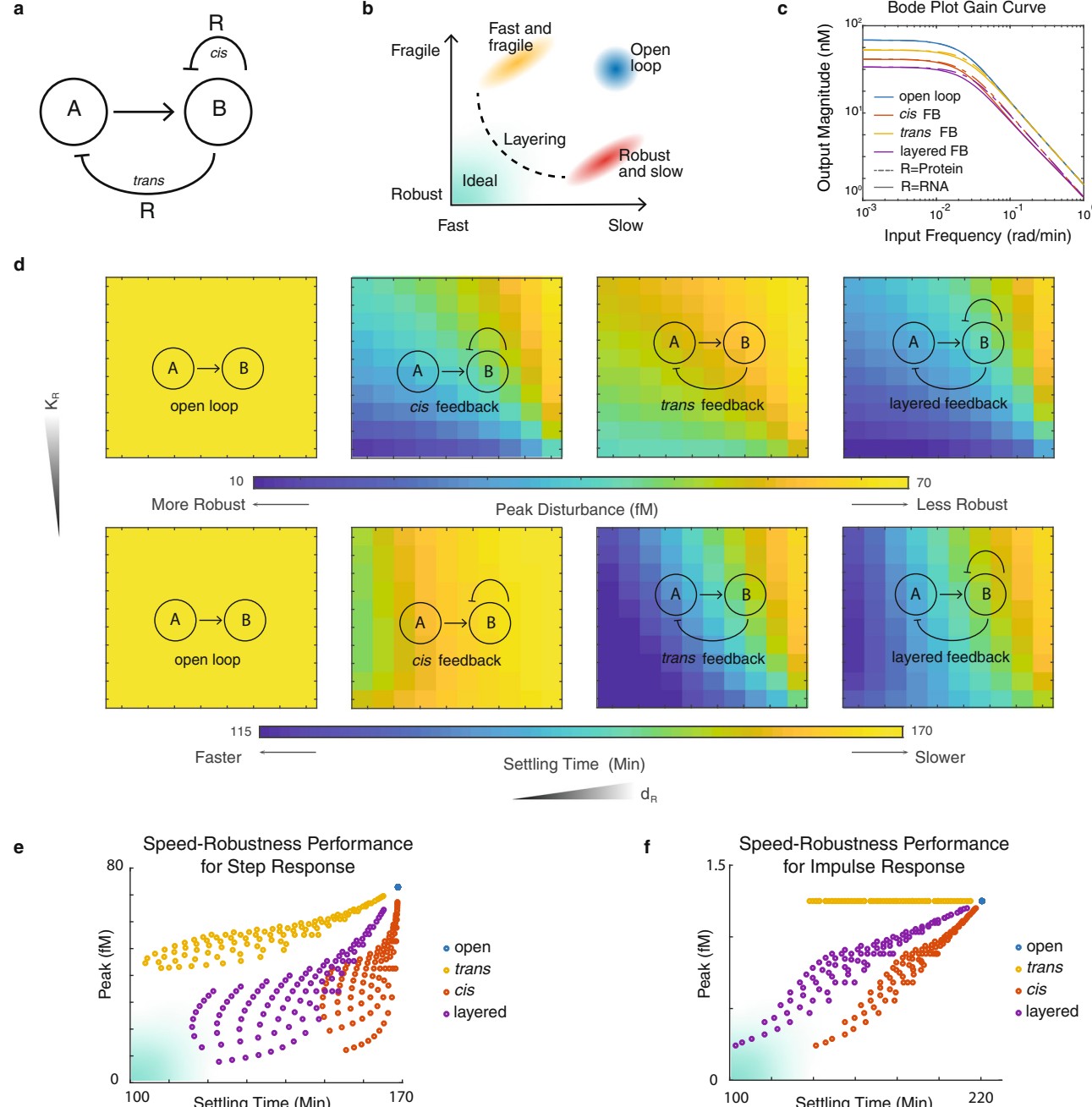

**Fig. 1 | Performance analysis of the layered feedback controller using a linearized state space model.** **a** Schematics of the node-based system. Here A and B are two molecular species, where A activates the expression of B and R. Functioning as a regulator, R regulates species B directly through the *cis* feedback on B and indirectly through the *trans* feedback on A. When both types of feedback exist in the same system, the configuration is termed layered feedback. **b** An illustration of the robustness-speed trade-off limit of feedback control. With a given set of parameters that define R, if one type of feedback is fast and fragile and another type of feedback is robust and slow, then layering these two feedbacks together could overcome the robustness-speed trade-off limit bound by either of these two feedbacks alone. **c** Response magnitude with production disturbance at different frequencies. This plot shows the magnitude response of species B (y-axis) when all three species' production rates are subjected to a perturbation (on $\gamma$) at different frequencies (x-axis). **d** Peak Disturbance and settling time on a step response in a two-dimensional parameter space defined by $K_R$ and $d_R$. The top panel shows the output peak disturbance when all three species' production rates are subjected to a step perturbation on $\gamma$. The bottom panel shows the time it takes for the systems to settle to a new equilibrium after the perturbation. The y-axis ($K_R$) represents the repression constant of regulator R; a large $K_R$ models a regulator with weak repression strength. The x-axis ($d_R$) represents the degradation rate of regulator R. **e** The four designs' step response performance evaluated in robustness and speed in a 10 × 10 parameter space. **f** The four designs' impulse response performance evaluated in robustness and speed in a 10 × 10 parameter space. Its corresponding heat maps are included in Supplementary Fig. 3.

consistent with what we learned from Fig. 1c. These results indicate that the two single-layer feedback controls are bound by a speed-robustness trade-off limit with a given regulator property, and layering could overcome that limit. Specifically, the *trans* feedback is faster but less robust; the *cis* feedback is slower but more robust. The layered feedback, on the other hand, appears to overcome this trade-off and achieve a better performance in the speed-robustness dimension in the same parameter space. Unsurprisingly, the performance of single-layer designs can be improved by reducing both parameters $K_R$ and $d_R$, which renders the desired regulator both strongly repressive and slow-

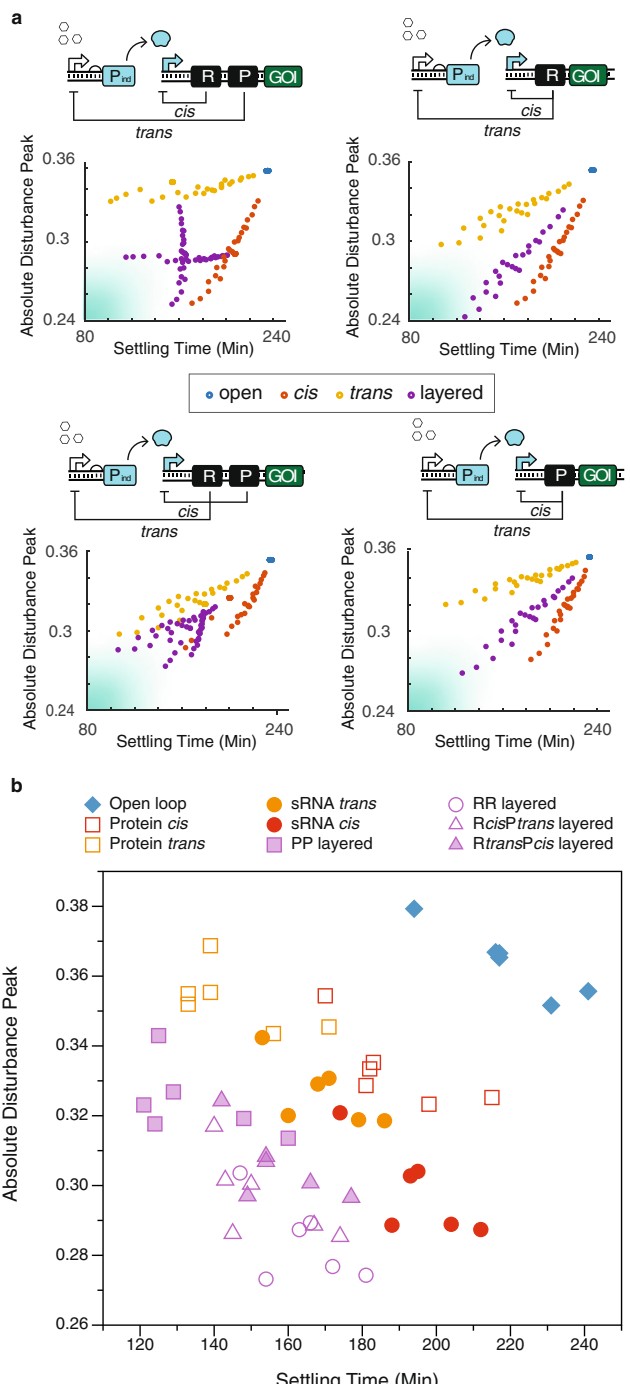

**Fig. 2 | Robustness-speed trade-off analysis on generic biomolecular configurations. a** The four possible biomolecular configurations and their simulated performance evaluated in robustness and speed with regulator parameter tuning. The four architectures are converted from the node-based design in Fig. 1a, with the regulator species being either sRNA or protein. In all cases, gene regulations happen at the transcriptional level. Each dot in the robustness-speed plots was computed by analyzing the simulated system dynamics with a 2-hour transcriptional perturbation at equilibrium. **b** The robustness-speed performance analysis of nine configurations. The x-axis represents the speed metric measured in settling time; the y-axis represents the robustness metric measured in absolute disturbance peak. The plot is generated from simulated dynamics by a 25% randomization on all model parameters. The simulated dynamics are shown in Supplementary Fig.4.

degrading. However, most transcriptional regulators found in nature are bound by the trade-off between binding affinity $K_R$ and degradation rate $d_R$. For instance, an antisense RNA can regulate transcription through RNA-RNA binding, which is highly repressive since this

## Table 1 | Generic Biomolecular Model Species

| Species | Description |
|---------|-------------|
| $M_{ind}$ | the mRNA of signaling protein $P_{ind}$ |
| $P_{ind}$ | the signaling protein translated peptides |
| $C_{ind}$ | the signaling complex, with folded signaling protein bound with inducer molecules |
| $R$ | the regulator sRNA repressor |
| $P$ | the regulator protein repressor |
| $M_G$ | the GOI mRNA transcript |
| $G$ | the translated GOI peptides |
| $G_m$ | the mature GOI (observable) |

regulation is often irreversible, but these RNA regulators are fast-degrading[23,24]. On the other hand, protein regulators degrade much slower, but their binding to DNA is often reversible. Hence, this trade-off raises a new design question: how do we choose the regulator R?

### Robustness-speed trade-off analysis with generic biomolecular configurations

To actuate the node-based design in the biomolecular context, we proposed four possible designs (Fig. 2a) with the regulator species being either R (regulatory small RNA, or sRNA) or P (regulatory protein). As shown in Fig. 2a, the system is induced with small molecule x, which activates the transcription of protein $P_{ind}$. Protein $P_{ind}$ sequentially activates the gene of interest (GOI) cassette, which contains gene GOI, R, and/or P. The four illustrations in Fig. 2a represent the four designs of layered feedback: (1) sRNA mediated *cis* feedback layered with protein mediated *trans* feedback, (2) sRNA mediated *cis* feedback layered with sRNA mediated *trans* feedback, (3) protein mediated *cis* feedback layered with sRNA mediated *trans* feedback, and (4) protein mediated *cis* feedback layered with protein mediated *trans* feedback.

Next, we interrogated the performance of these four designs in simulation with low frequency pulse disturbances. The dynamics of these systems are described with the reduced differential equations in Eqs. (4)–(11). The detailed species and parameters involved in this model are listed in Tables 1 and 2, respectively. The parameters were estimated based on previous parameterization work[25] or calculated with values obtained from BioNumbers[26] (see SI). It is worth noting that we purposefully chose the degradation rates ($d_r$, $d$) and repression coefficients for regulatory RNA $R$ and regulatory protein $P$ ($K_R$, $K_p$) to simulate the same effective repressive strength for fair comparison. The regulatory RNA was simulated to have strong binding affinity (small $K_R$) but is quickly degraded (large $d_r$); the regulatory protein was simulated to have weak binding affinity (large $K_p$) but is slowly degraded (small $d$).

$$\frac{dM_{ind}}{dt} = f_{tx} \cdot \beta_A \cdot \left(\frac{x}{K_x + x}\right) \cdot f_{trans} - d_m \cdot M_{ind} \quad (4)$$

$$\frac{dP_{ind}}{dt} = f_{tl} \cdot k_{tp} \cdot M_{ind} - d \cdot P_{ind} - K_r \cdot P_{ind} \quad (5)$$

$$\frac{dC_{ind}}{dt} = K_r \cdot P_{ind} - d \cdot C_{ind} \quad (6)$$

$$\frac{dM_G}{dt} = f_{tx} \cdot \beta_B \cdot \left(\frac{C_{ind}}{K_{ind} + C_{ind}}\right) \cdot f_{cis} - d_m \cdot M_G \quad (7)$$

$$\frac{dR}{dt} = f_{tx} \cdot \beta_B \cdot \left(\frac{C_{ind}}{K_{ind} + C_{ind}}\right) \cdot f_{cis} - d_r \cdot R \quad (8)$$

**Table 2 | Generic Biomolecular Model Parameters**

| Parameters | Description | Unit | Estimates |
|---|---|---|---|
| $\beta_A$ | max transcription rate of the inducible promoter $P_x$ | fM/min | 2 |
| $K_x$ | activation coefficient of the chemical inducer $x$ | nM | 1.4e4 |
| $K_R$ | repression coefficient of the sRNA repressor | fM | 20 |
| $K_p$ | repression coefficient of the protein repressor | fM | 200 |
| $d_m$ | degradation/dilution rate of mRNA | 1/min | 0.1 |
| $k_{tp}$ | translation rate of the inducing protein $P_{ind}$ | 1/min | 0.1 |
| $d$ | degradation/dilution rate of proteins, dominated by dilution | 1/min | 0.03 |
| $K_r$ | the maturation rate of the activating complex | 1/min | 0.1 |
| $\beta_B$ | max transcription rate of the $C_{ind}$ inducible promoter | fM/min | 20 |
| $K_{ind}$ | activation coefficient of $C_{ind}$ | nM | 200 |
| $d_r$ | degradation/dilution rate of sRNA | 1/min | 0.3 |
| $\alpha$ | maturation rate of GOI | 1/min | 0.2 |
| $x$ | the chemical inducer that activates $P_x$ | nM | 2.0e6 |
| $k_{tg}$ | translation rate of GOI | 1/min | 0.1 |
| $f_{tx}$ | scaling factor of universal transcription | NA | 1 |
| $f_{tl}$ | scaling factor of universal translation | NA | 1 |
| $k_{tr}$ | translation rate of the regulator protein | 1/min | 0.04 |
| $d_{pr}$ | degradation/dilution rate of regulator proteins, dominated by dilution | 1/min | 0.03 |

$$\frac{dP}{dt} = f_{tl} \cdot k_{tr} \cdot M_G - d_{pr} \cdot P \tag{9}$$

$$\frac{dG}{dt} = f_{tl} \cdot k_{tg} \cdot M_G - \alpha \cdot G - d \cdot G \tag{10}$$

$$\frac{dG_m}{dt} = \alpha \cdot G - d \cdot G_m \tag{11}$$

If transcription is regulated by a protein species, the Hill function is written as $f_{Hill_P} = \frac{K_p}{K_p + P}$. If transcription is regulated by a sRNA species, the Hill function is written as $f_{Hill_R} = \frac{K_R}{K_R + R}$. For the open loop, $f_{cis} = f_{trans} = 1$. For the *trans* only feedback, $f_{cis} = 1$ and $f_{trans} = f_{Hill_P}$, if mediated by a protein species; $f_{trans} = f_{Hill_R}$, if mediated by a sRNA species. For the *cis* feedback, $f_{trans} = 1$ and $f_{cis} = f_{Hill_P}$, if mediated by a protein species; $f_{cis} = f_{Hill_R}$, if mediated by a sRNA species. For the layered feedbacks, the four possible combinations are: (1) R *cis* R *trans*, $f_{cis} = f_{Hill_R}$ $f_{trans} = f_{Hill_R}$; (2) R *cis* P *trans*, $f_{cis} = f_{Hill_R}$ $f_{trans} = f_{Hill_P}$; (3) P *cis* R *trans*, $f_{cis} = f_{Hill_P}$ $f_{trans} = f_{Hill_R}$; (4) P *cis* P *trans*, $f_{cis} = f_{Hill_P}$ $f_{trans} = f_{Hill_P}$.

We simulated the system dynamics of all four sets of designs (each with open, *cis*, *trans*, and layered feedback) and scaled individual trajectories by their equilibrium values to allow fair comparison. We gave the universal transcriptional rate scalar $f_{tx}$ a 25% impulse drop for 120 min at equilibrium. Then we analyzed the simulated dynamical profile to determine the system's robustness and speed by identifying its absolute disturbance peak and settling time. In Fig. 2a, we scanned the parameter space for both the RNA and protein regulators. The model used for simulation is listed in Eqs. 4 to 11. The parameter search covers a $5 \times 5$ space on the binding affinity and degradation rate of the regulators while all other parameters are held constant, as shown in

Table 2. Specifically, we search R with $K_R$ ranging from 20 fM to 100 fM and $d_r$ ranging from 0.06/min to 0.3/min; then we searched P with $K_p$ ranged from 40 fM to 200 fM and $d_{pr}$ ranging from 0.03/min to 0.15/min. We observed a similar pattern as the parameter scan with the linearized state space model, as shown in Fig. 1. In comparison, the *trans* feedback is faster but more fragile, the *cis* feedback is slower but more robust. Additionally, tuning the regulators does improve the performance of the two single-layer designs, but layering appears to reach further into the optimal performance space of robustness and speed.

Finally, for each of the nine configurations, we used six sets of parameters to simulate six dynamical profiles in response to perturbations. These parameter sets were randomized within a 25% perturbation on the full parameter set listed in Table 2 to emulate biological uncertainties. The settling times and the absolute disturbance peak of each simulation were recorded in the robustness-speed performance plot in Fig. 2b. Here, we also observed a similar performance pattern as the results shown in Figs. 1 and 2a. To move forward, we chose the design with sRNA-mediated *cis* feedback and protein-mediated *trans* feedback to proceed to the experimental portion of the work. Because in addition to demonstrating superior robustness-speed performance, it also diversifies the architecture with both protein and RNA regulators.

## Experimental implementation of a layered feedback control in living E. coli cells

We designed four genetic constructs to test and experimentally analyze the layered feedback design we proposed based on the simulated results in Fig. 2b. We aimed to compare the dynamical performances of the open loop, the *cis* feedback, the *trans* feedback, and the layered feedback. With the effect of genetic context in mind[27], we designed the four constructs with the same configuration. As shown in Fig. 3a, the system is composed of two cassettes expressed on two plasmids. The combinatorial promoter $P_{Rhl/LacO}$[28] drives the expression of CinR on a medium copy p15A backbone; promoter $P_{cin}$ drives another cassette containing sRNA repressor pair (AS and Att)[24], sfYFP, and LacI expressed on a high copy ColE1 backbone. The system is activated by the induction of a combinatorial promoter $P_{Rhl/LacO}$ with an AHL inducer Rhl. The promoter then allows the expression of protein CinR, which subsequently binds to an AHL inducer Cin in growth medium to activate promoter $P_{cin}$. The promoter $P_{cin}$ controls the transcript that contains sRNA regulating pair AS and Att, the observable system output sfYFP, and the protein regulator LacI. In this configuration, sRNA repressor AS regulates the expression of sfYFP via the *cis* feedback, and LacI represses $P_{Rhl/LacO}$ to regulate the expression of CinR through the *trans* feedback. To avoid genetic context change and metabolic load variation, we created mutated regulator pieces to disable feedbacks without changing the genetic context. Specifically, we paired AS with its orthogonal attenuator Att(M)[29] to disable the *cis* feedback, and we built LacI(M) off LacI with the LacO binding site sequence removed to disable the *trans* feedback. The construction of these four genetic networks is a non-trivial task. It involves one combinatorial regulation, one activation cascade, and two nested autoregulatory motifs. Extensive part-characterization and expression optimization is required to confirm the networks' proper dynamical properties.

To characterize all the parts and confirm their functionalities, we started with the promoter $P_{Rhl/LacO}$. Because LacI was designed as a regulatory part of the construct, we chose strain JS006[30] for its ΔLacI genotype and integrated a constitutively expressed RhlR into its genome. We characterized the promoter $P_{Rhl/LacO}$ in this RhlR containing JS006 strain with 200 $\mu M$ Rhl induction (Fig. 3b). The test confirmed that the combinatorial promoter has sufficient fold induction with about 5% leaky expression. Then, we tested the sRNA mediated *cis* feedback with AS-Att pair and AS-Att(M) pair. We confirmed that the AS-Att(M) pair functions as an appropriate control for the *cis* feedback, as the dynamics of the two constructs are consistent with previous

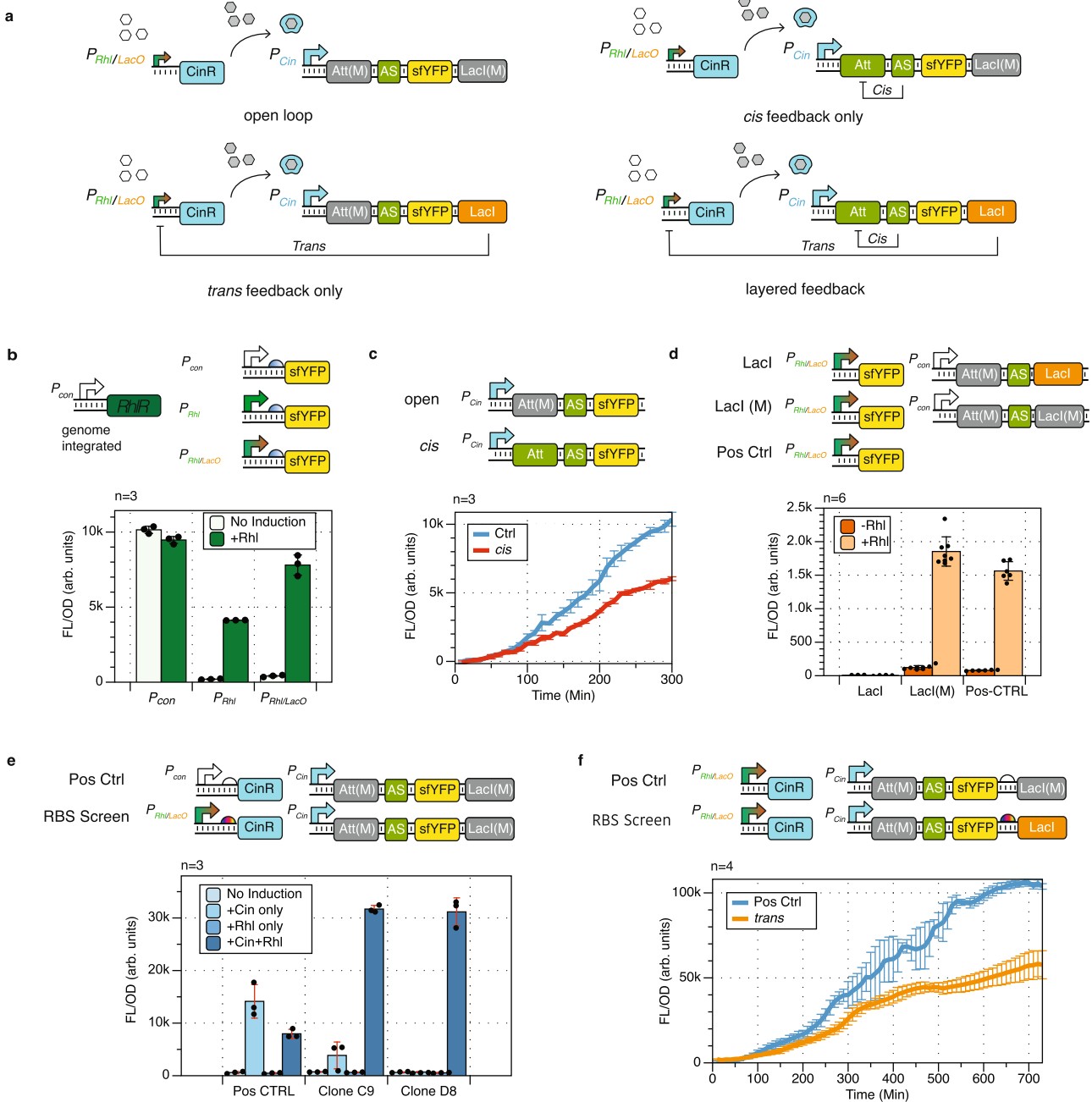

**Fig. 3 | Experimental construction of the layered feedback control in E. coli.**
**a** The four genetic constructs designed for experimental construction of the open loop, the *cis* feedback, the *trans* feedback, and the layered feedback. **b** Part-characterization of the combinatorial promoter $P_{Rhl/LacO}$ on its activation function. The signal output of $P_{Rhl/LacO}$ upon Rhl induction, compared with the constitutive promoter $P_{con}$ and the Rhl-inducible promoter $P_{Rhl}$, performing in an RhlR integrated JS006 strain. **c** Part-characterization of sRNA regulator AS with its paired attenuator Att and its orthogonal attenuator Att(M). **d** Part-characterization of the combinatorial promoter $P_{Rhl/LacO}$ on its repression function, tested with its

repressor LacI and a mutated repressor LacI(M). LacI(M) was built off LacI, with the LacO binding site deleted. **e** RBS screening results for the expression of CinR to enable activation cascade. The RBS in Clone D8 was chosen to proceed with the following stages of construction. **f** RBS screening result for the expression of LacI to facilitate the *trans* feedback. The plot shows that the selected clone confirmed the functionality of the negative feedback without over-repression. All data are presented as mean value +/- standard deviation of n samples, *n* = number of biological replicates.

findings[31] (Fig. 3c). Next, we examined the repressive regulatory function of combinatorial promoter $P_{Rhl/LacO}$ with LacI and LacI(M) (Fig. 3d). The result confirmed that $P_{Rhl/LacO}$ can be fully repressed by LacI with full Rhl induction. In the meantime, we determined that LacI(M) is an appropriate control for the *trans* feedback.

At last, we performed two sets of ribosome binding site (RBS) screening assays to achieve the ideal protein expression strength that actuates the desired circuit functionalities. First, we searched for an

RBS to express CinR so that it facilitates the activation cascade in our circuit design context. The cascade was designed to be turned on by the inducible expression of CinR through promoter $P_{Rhl/LacO}$. Subsequently, CinR binds to inducer Cin in the growth medium, and then the CinR-Cin complex activates the promoter $P_{cin}$. Because $P_{Rhl/LacO}$ exhibited some leaky expression, the RBS for CinR required optimization. If the RBS is too weak, the downstream cascade could be under-activated; if the RBS is too strong, the leaky expression of $P_{Rhl/LacO}$

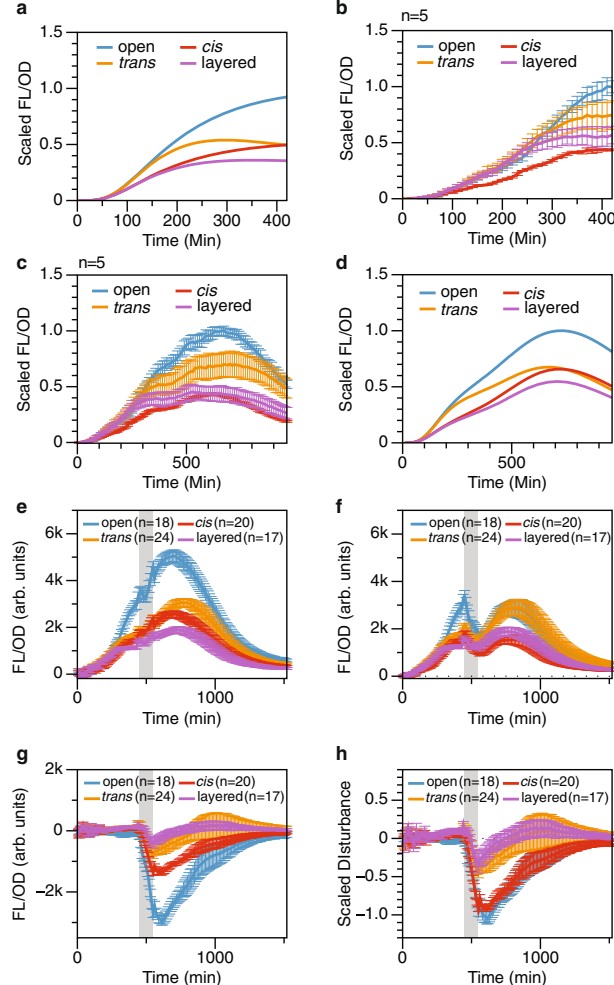

**Fig. 4 | Dynamics of the four testing constructs – the open loop, the *cis* feedback, the *trans* feedback, and the layered feedback, as shown in Fig. 3a.**
**a** Simulated dynamics of the four constructs with the reduced model in Eqs. (4)–(11) in a 7-hour window. **b** Experimentally observed dynamics of the four constructs in a 7-hour window. **c** Experimentally observed dynamics of the four constructs in a 16-hour window. **d** Simulated dynamics of the four constructs with a growth-dependent model in Supplementary Section 1 in a 16-hour window. **e–h** Dynamic analysis of the test and intact groups in the AHL-Rhl and AHL-Cin wash perturbation experiment. Grey regions indicate the duration of disturbances. **e** Dynamical profile of the intact group. **f** Dynamical profile of the test group. **g** Disturbance profile, calculated by subtracting the dynamics of the test group from the dynamics of the intact group. **h** Scaled disturbance profile. The dynamical deviation of the test group from the intact group is scaled by each construct's pre-disturbance output value. All data are presented as mean value +/− standard deviation of *n* samples, *n* = number of biological replicates.

## Robustness and speed under transcriptional perturbations evaluated in living E. coli cells

After building the four testing constructs in Fig. 3a, we sought to confirm whether our constructs were properly built by comparing measured dynamical profiles with simulated dynamics. Based on the simulated dynamical profile we generated with the reduced model (Fig. 4a, Eqs. 4–11), we expected the *trans*, the *cis*, and the layered feedback constructs to have lower equilibrium output signals and faster responses after the induction of Rhl and Cin at $t = 0$. We indeed observed this profile experimentally in the first 7 hours of our dynamical experiment (Fig. 4b). However, as the experiment continued to 16 h, we observed that the reduced model no longer qualitatively describes the observed dynamics (Fig. 4c). Notably, the experimental dynamics no longer reach equilibrium. Is this discrepancy due to errors in circuit construction or in dynamical models?

We know that the reduced model was built with numerous assumptions that do not hold as a bacterial culture grows towards stationary phase. One of many affected parameters is protein dilution $d$, which is dominated by cell division. The reduced model assumes a constant cell division rate, therefore a constant $d$. However, we observed that the culture grew beyond the exponential phase and started to slow down as the population reached the stationary phase (Supplementary Fig. 2). As cell division slows down, protein dilution slows down, but the active protein degradation due to nutrient starvation speeds up[32]. At the same time, protein and mRNA production also slow down as they are tied to the nutrient availability in the growth medium[33,34]. We then expanded our model with functions that approximate the connection between gene expression, cell growth, and nutrient availability (see Supplementary Section 1). The dynamical profile we obtained (Fig. 4d) roughly resembles the observation in Fig. 4c, which confirmed the functionality of our constructs beyond the exponential phase. Although much more work is needed to create an accurate multiscale model with identified parameters, this preliminary model provides insights for understanding the gene expression dynamics across multiple growth phases.

Since the system dynamics in living cells do not reach equilibrium, we cannot use the same method in Fig. 2 to interrogate system dynamics. To overcome this problem, we designed each perturbation experiment with a test group and an intact group. Figure 4e, f show the two groups from one of the six sets of perturbation experiments. This experiment was designed to emulate the simulated results in Figs. 1 and 2, where the impulse perturbation is only applied on the transcriptional rate of the synthetic system. Specifically, both groups were induced with 100 µL of Rhl and 10 µL of Cin at $t = 0$ and let grow for 7 h. At hour 7, both groups were washed with phosphate buffered saline (PBS), and the growth medium was replaced. The intact group shown in Fig. 4e was resuspended in fresh medium with 100 µL of Rhl and 10 µL of Cin; the test group was resuspended in fresh medium with no inducer. The perturbation window was 120 min (greyed section in Fig. 4e–h), then both groups were washed and resuspended back to the recycled pre-disturbance media. We measured the disturbance of constructs by the deviation of the test group dynamics from the intact group dynamics (Fig. 4g). Since the four constructs did not have the same signal output level, we scaled the disturbance profile in Fig. 4g by each construct's pre-disturbance output value to obtain a fair comparison (Fig. 4h). This method allowed us to access the dynamical profiles of synthetic biological networks under perturbations without a system equilibrium. It is also easily applicable to future studies of system dynamics in biology.

In Fig. 5, we showed the performance analysis of the four architectures with the open loop, the *trans* feedback, the *cis* feedback, and the layered feedback control from the scaled disturbance profile in Fig. 4h. The experiment was performed with 20 replicates for the open-loop construct and 24 replicates for each of the three feedback-controlled constructs. The cells that demonstrated atypical growth

could cause the cascade to be over-activated without Rhl induction (Supplementary Fig. 1a). Here, we screened for an RBS so that the system is off (shows minimal sfYFP signal) with no inducer, only the Rhl inducer, or only the Cin inducer. Then the system turns on with both Rhl and Cin inducers. As shown in Fig. 3e, Clone D8 presented the desired induction profile. The resulting RBS was selected to proceed with the circuit construction. Finally, we screened the RBS for LacI so that it facilitates the *trans* feedback. As we see in Fig. 3d, LacI has a strong repressive effect on $P_{Rhl/LacO}$ with full Rhl induction (Supplementary Fig. 1a). Therefore, a relatively weak RBS is required for the *trans* feedback to function (Supplementary Fig. 1b). Figure 3f shows the desired dynamical profile of a clone resulting from the screen, where LacI is expressed enough to present a repressed trajectory yet is not over-expressed to shut off the sfYFP expression entirely.

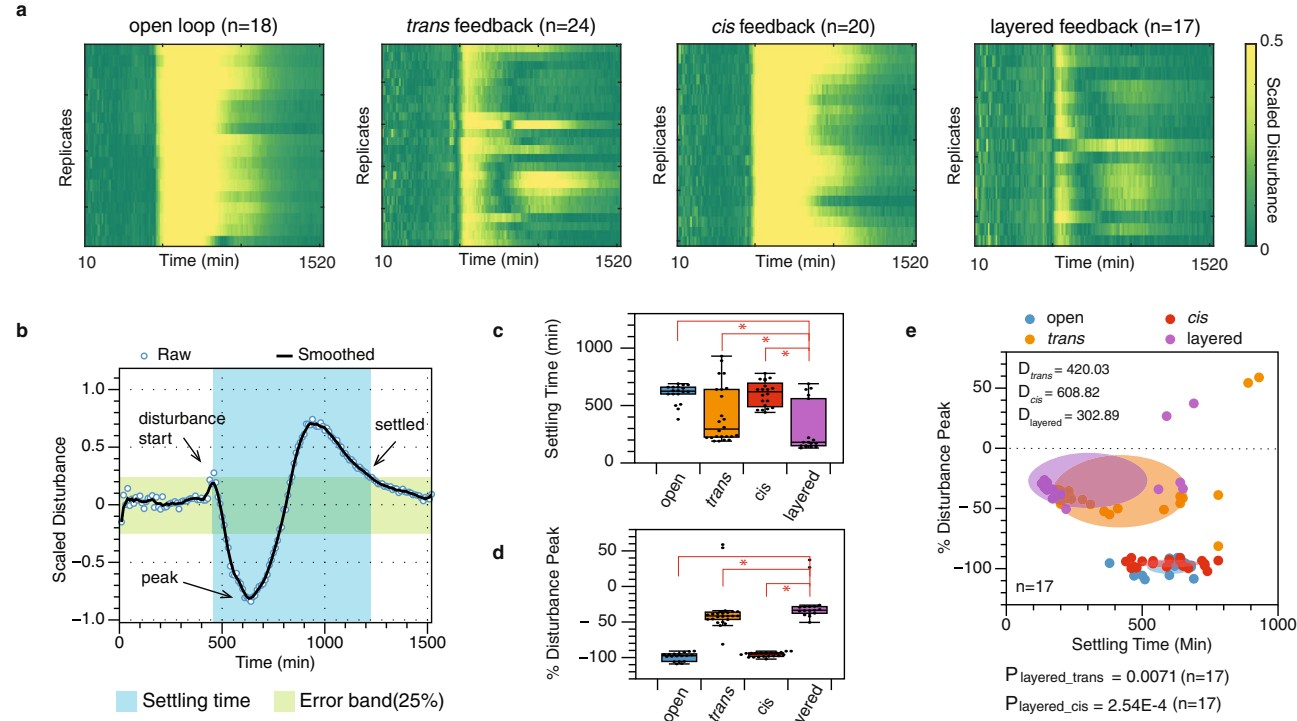

**Fig. 5 | Performance analysis of the four constructs with the open loop, the *trans*, the *cis*, and the layered feedback controls in living E. coli cells with an impulse transcriptional perturbation caused by inducer wash. a** Dynamical profile of four network architectures with n data traces, where *n* = number of biological replicates. In an individual heat map, each row represents the disturbance dynamics of each liquid culture well. Each column represents the measurements at a given time point. The color intensity indicates the absolute magnitude of output caused by the input disturbance. **b** The metrics used to extrapolate speed and robustness from experimental data. **c** Speed comparison of the four network designs measured in settling time. **d** Robustness comparison of the four network designs measured in percentage disturbance peak. Data in **c**, **d** are presented with standard box plots overlaying all individual data points. Box plots show centre line as median, box limits as upper and lower quartiles, whiskers as minimum to

maximum values. The asterisks denote statistical significance between the two groups, determined by paired one-tail Student's *t*-tests. Additional statistical information is listed in Supplementary Table 8. **e** The robustness-speed performance of the four constructs under inducer chemical wash perturbation is presented in trade-off plots. Each dot renders the robustness-speed performance of individual cell culture; each shaded area represents the robustness-speed performance within one standard deviation. The distance of each point to the origin quantifies the robustness-speed performance of an individual test culture. The *D*-values denote the average performance of its corresponding group. The value $P_{layered-trans}$ denotes the p-value between the layered control and the *trans* control, and $P_{layered-cis}$ denotes the p-value between the layered control and the *cis* control. p-values are determined by the paired one-tail Student' *t*-test with n samples. *n* = number of biologically independent replicates.

dynamics would be excluded in data analysis (see Supplementary Fig. 5). First, we generated heat maps to illustrate the dynamical profiles of individual replicates to showcase the system performance in the context of biological uncertainty. In Fig. 5a, each heat map represents the output disturbance profile of all remaining replicates after data exclusion. They reveal the patterns of systems' output disturbance after the chemical perturbation was introduced at steady-state and then removed after two hours. In these four heat maps, we see that the *trans* feedback is both more robust (dimmer at the peak) and faster (dims down faster) compared to the *cis* feedback. The layered feedback also appears to be the most robust and speedy. Then, we analyzed each replicate's trajectory to quantify this result, as shown in Fig. 5b. We used the scaled peak disturbance value as the metric for robustness and the system's settling time as the metric for speed. The blue circles represent the scaled disturbance profile of a single colony taken from Fig. 4h. The solid line is a smoothed curve estimated from the experimental data using the MATLAB *rlowess* function. The blue time zone is the settling time zone calculated from the beginning of the perturbation to the time it takes for the profile to recover into the error band. The error band in this experiment is defined to be 0.25. As marked in the plot, the peak disturbance is the furthest point away from the *x*-axis. Then we showed the comparison of these two metrics across the four construct designs (Fig. 5c, d). The result was consistent with the visual representation in Fig. 5a, where the *trans* feedback is both more robust and rapid compared to the *cis* feedback. The layered feedback shows the most optimal

performance in robustness and speed compared to the open-loop and the single-layer constructs. The differences were statistically significant. Finally, we mapped these two metrics into the trade-off plot as shown in Figs. 1a and 2c. The trade-off plot portrays the robustness and speed of individual cell cultures in this experiment. The distance of each point from the origin quantifies the dynamical performance of each replicate measured in robustness and speed. The figure showed that the average robustness-speed performance of the layered feedback construct (purple shaded area) was superior to the average performances of the *trans* feedback (orange shaded area) and the *cis* feedback (red shaded area); the differences were statistically significant (p-values were shown in Fig. 5e). Interestingly, we noticed that the *cis* feedback did not appear to improve the system performance from the open loop. However, despite its lacking of effectiveness as a single-layer controller, layering it with the *trans* feedback still granted the system further improvement.

## Interrogating controller performance under complex perturbations

In electrical engineering and control theory, a system's response to disturbances is usually evaluated with a single perturbed input. However, in biological systems, perturbations are often applied at the environmental level, which could have much broader, yet poorly understood impacts on the system dynamics. For instance, the temperature and nutrient availability fluctuations in natural environments cause physiological stress responses in cells to ensure adaptation and

survival. As a consequence, the dynamics of synthetic systems encoded in the cells would be severely impacted as they are integrated as part of the inner cellular network. Additionally, our generic models are usually over-simplified for simplicity. They often omit many physiological constraints in the inner cellular environment. One example is the translational capacity, which is modeled as a constant rate disregarding the limitation of translation resources. As one of the most costly processes in cells, translation is limited by resource availability. It is documented by previous work that this resource limitation causes retroactivity in synthetic circuits[35]. In this section, we performed five additional experiments to interrogate feedback controls' effects and limitations under poorly understood perturbations (Fig. 6).

In our generic biomolecular design (Fig. 2) and the first set of perturbation experiments (Fig. 4 and 5, perturbations were introduced as a universal low-frequency transcriptional impulse drop. Here, we applied the same transcriptional perturbation by manipulating the concentration of Cin and Rhl in the positive direction (Fig. 6a). We observed that, in this perturbation experiment the open-loop appeared to have the best performance, which was not predicted by the generic model in Eqs. (4)–(11) (Fig. 6a, S6). We hypothesize that this is because the disturbance on the open loop is naturally attenuated by translational resource capping. We used a simple model in Supplementary Section 1.3 and Supplementary Fig. 6 to show that if the translation rate slows down as the system accumulates too much mRNA, the open loop shows a natural attenuation during a transcriptional spike. This is because the open loop has more mRNA at equilibrium than the constructs with feedback controls. The increasing transcriptional rate would produce even more mRNA, but it would also further slow down the translational rate to negate this effect. Nevertheless, the remaining controlled constructs showed that the layered feedback control inherited performance characteristics from the two single-layer constructs. More importantly, the layered control appeared to outperform the single-layer feedback with statistical significance.

Then, in the temperature perturbations that emulate one of the most common environmental disturbances in biological systems (Fig. 6b, c, and Supplementary Fig. 7), we perturbed the system by dropping the incubation temperature from 37 °C to 30 °C (Fig. 6b) and spiked it with a temperature increase from 37 °C to 42 °C (Fig. 6c). It was unexpected that the response profiles of the *trans* feedback and the *cis* feedback in these two experiments demonstrated opposite traits. Notably, when the temperature dropped, the *trans* feedback dynamics shifted down, and the *cis* feedback dynamics shifted up (Fig. 6b), with the *cis* feedback out-performing the *trans* feedback. When the temperature spiked, the *trans* feedback dynamics shifted up and the *cis* feedback dynamics shifted down (Fig. 6c), with the *trans* feedback out-performing the *cis* feedback. It is unclear what parameters are perturbed with temperature fluctuations because the perturbation is effectively applied to living cells' physiological states. However, since all constructs were engineered to ensure consistent context, this pattern is most likely caused by the property changes of the regulator sRNA and regulator protein. We deduced that, since the open loop in both cases shows disturbance in the negative direction, there is most likely a transcriptional downshift when the temperature shifts away from the optimal 37 °C. In addition, the pattern suggests that the sRNA repression is more severely weakened at 30 °C, and the protein repression is more severely weakened at 42 °C. In Fig. 6c with a temperature up-shift from 37 °C to 42 °C, the *trans* feedback dynamics upshift could be explained by a faster denaturation of proteins at 42 °C and a weakened protein-DNA binding affinity. Yet, at 30 °C (Fig. 6b), there is no reason for sRNA to be degraded faster at a lower temperature. Hence, we hypothesized that the weakening in sRNA could be due to a less efficient maturation step. Previous work found that there might be a folding delay for PT181 sRNA to mature to its functional form[25]. We suspect that a lower temperature could cause sRNA

to misfold at a higher rate due to increased tolerance to mispairing when temperature decreases. This misfolding could also cause a higher failure rate to terminate transcription at the antisense-attenuator binding site, hence weakening the RNA-RNA binding-induced transcriptional repression. We used a simple model in Supplementary Section 1.2 and Supplementary Fig. 7 to describe this hypothesis. The simulated dynamics qualitatively agreed with the experimental observation. It is worth noting that, with both upshift and downshift temperature perturbation experiments, the layered feedback appears to mitigate the opposite effects of the two single layers. Although in the temperature spike experiment, the *trans* feedback design outperforms the layered feedback with its fast settling time, the layered feedback still outperforms the *cis* feedback design (Fig. 6c and Supplementary Fig. 7). Overall, this set of experiments suggests that layering feedback controls with both sRNA and protein regulators is an effective performance optimization strategy for synthetic biomolecular networks in environments subjected to temperature fluctuations.

One other type of common environmental perturbation is the fluctuation of the nutritional resource. In Fig. 6d, e, we tested the system's dynamics when subjected to a glucose perturbation in both directions. Specifically, in Fig. 6d, we temporarily increased the glucose concentration from 0.1% to 1% for two hours. In Fig. 6e, we temporarily decreased the glucose concentration from 1% to 0% for the same duration. Interestingly, while we were expecting a signal drop with temporary glucose starvation, we observed that all four constructs showed a slight dip and then an upshift in the output signal that was amplified over time, even after being restored to the 1% glucose media. We deduced that, since neither open loop dynamics in Fig. 6d and e trend towards recovery, this impulse perturbation appears to cause an upward step disturbance in the translational rate of all proteins in the system, as the translational processes heavily consume the carbon source. This could explain the observation in Fig. 6d with a temporary increase in glucose availability, which could increase the amino acid abundance in the cells. However, the drastic dynamical upshift observed in Fig. 6e also suggests the same upward step disturbance in the translational rate. The bacterial stringent response is known to react to nutrient downshift to down-regulate both transcription and translation[36]. Therefore, it is unclear why glucose starvation would cause a drastic long-lasting dynamical upshift. Nevertheless, we used a simple model (Supplementary Section 1.3 and Supplementary Fig. 8) to describe our hypothesis. The resulting simulated dynamics showed that the model is plausible, given the observation. Since most of the trajectories were not trending towards recovery after 26 h of growth, we did not compute the settling time for this experiment. On the other hand, the computed robustness metric indicated that the *trans* feedback and the layered feedback constructs were the most robust against this perturbation. Therefore, it is unclear why the temporary nutrient down-shift would cause such a drastic dynamical up-shift.

## Discussions

This work explored the feasibility of layered feedback as a control optimization strategy in biomolecular networks both in silico and in living cells. We first showed that the layered architecture overcomes the robustness-speed trade-off in a wide parameter space with a node-based linearized state space model. Then, we expanded the design to a generic biomolecular model to simulate system dynamics with perturbations. With the guidance of theory and simulation, we forward engineered a layered feedback controlled synthetic network in living E. coli cells. We successfully constructed this network with a combinatorial regulation, an activating cascade, and two nested auto-regulatory motifs to achieve its desired dynamical functions. Finally, we performed six dynamical experiments with chemical, temperature, and nutrient perturbations to access the disturbance profiles of these

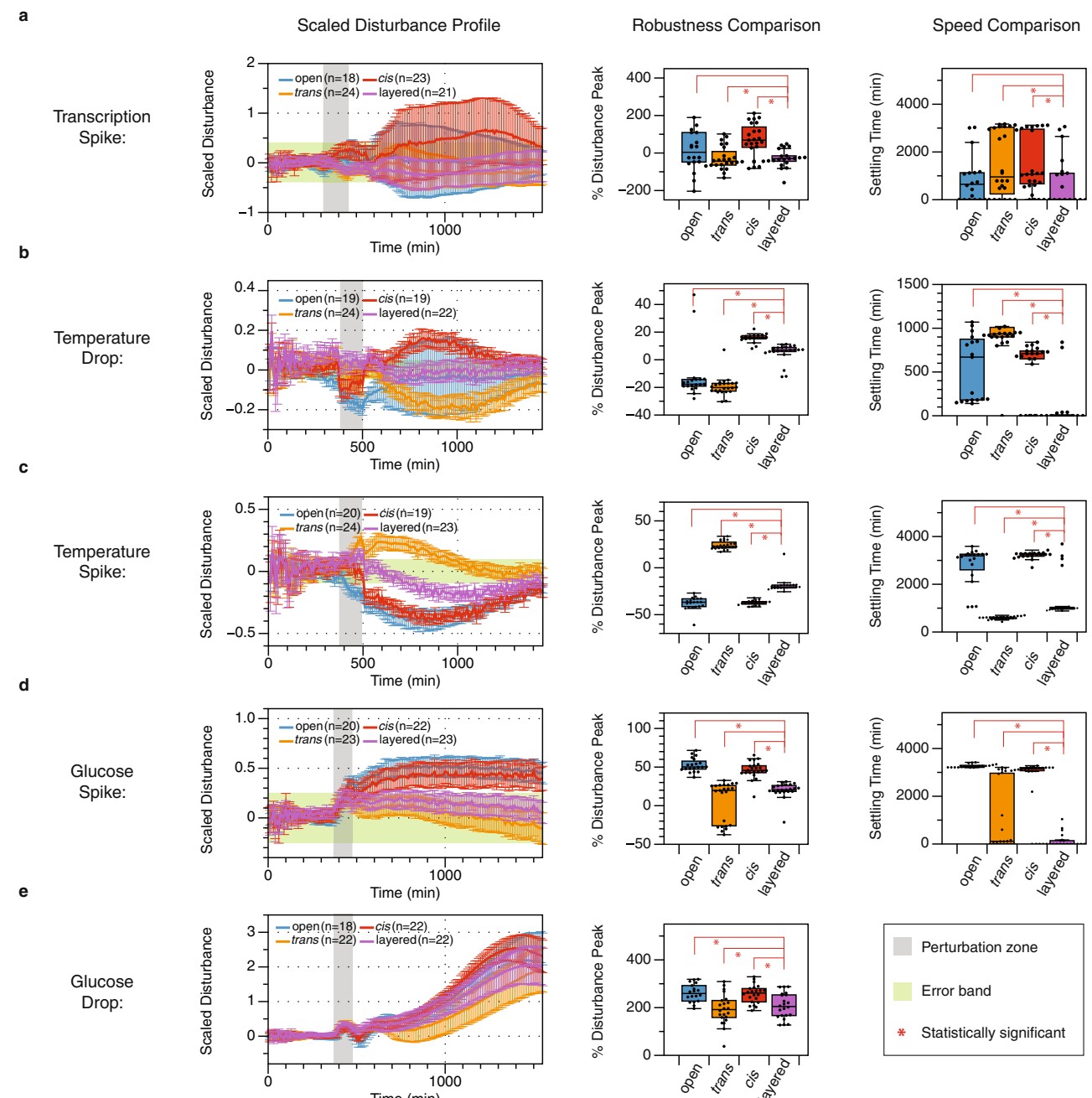

**Fig. 6 | Interrogating controller performance under complex perturbations.**
The first column presents the scaled disturbance profile, which is the dynamical deviation of the test group from the intact group, scaled by each construct's pre-disturbance output value. Data in this column are presented as mean value +/− standard deviation of $n$ samples, $n$ = number of biological replicates. The second and third columns present the robustness and the speed of the four network designs. Data from these two columns are presented with standard box plots overlaying all individual data points. Box plots show centre line as median, box limits as upper and lower quartiles, whiskers as minimum to maximum values. The asterisks denote statistical significance between the two groups, determined by paired one-tail Student's t-tests, additional statistical information is listed in Supplementary Table 8. If a trajectory does not recover back to the error band at the end of the measurement, a high-order polynomial function was then used to fit the data and predict its settling time. The asterisks denote statistical significance between the two groups. **a** System dynamics under chemical perturbation (spike), where the inducers AHL-Cin increased from 3 μM to 10 μM, and the AHL-Rhl increased from 20 μM to 100 μM at 310 min. Subsequently, the cultures were restored to the pre-disturbance media after 2.5 h. **b** System dynamics under temperature perturbation (dip). The temperature drops from 37 °C to 30 °C at 380 min and restores to 37 °C at 510 min. **c** System dynamics under temperature perturbation (spike). The temperature increases from 37 °C to 42 °C at 380 min and restores to 37 °C at 510 min. **d** System dynamics under nutrient perturbation (spike). The glucose concentration spikes from 0.1% to 1% at 370 min and restores to 0.1% at 500 min. **e** System dynamics under nutrient perturbation (dip). The glucose concentration drops from 1% to 0% at 370 min and restores to 1% at 500 min.

synthetic networks. We found that, as predicted by the model, the layered feedback control outperforms the single-layer feedbacks in robustness and speed when under chemical perturbations. In the temperature perturbation experiments, the layered feedback

neutralizes the opposite responses of single-layer constructs to stabilize the systems. In the nutrient perturbation experiments, although not all systems recovered, we observed the optimization effect of the layered control in disturbance attenuation. These results not only

provide validated guidance for controller design for synthetic biological networks, they also offer insights for understanding nature's dynamical control strategies.

Through the measurement and analysis of all six sets of experiments, many aspects of the dynamics posed interesting questions regarding natural biological networks. These observations suggested that pieces in our synthetic circuits were coupled through unknown mechanisms that await discoveries, which provided new directions for future works. Nevertheless, without knowing all the mechanistic details, we were able to use effective models to guide the design and construction of a layered feedback controller in synthetic biological networks. The experimentally observed dynamics also confirmed the model's prediction on the layered architecture's performance optimization effects.

The experimentally observed gene expression dynamics across different growth phases also revealed the dynamical entanglement across the populational and the molecular scales. We found that the common assumptions on constant cell division and unlimited resources are responsible for the major discrepancies between simulated and observed system dynamics. For system modeling, the quality of a model is bound by the accuracy and identifiability trade-off. Reduced models have fewer species and parameters so that they are more identifiable but less accurate and relevant due to oversimplification. On the other hand, chemical reaction network models are more accurate with fewer assumptions, but they are usually substantial in size and difficult to system identify. For future studies, we plan to layer the two types of models to overcome this trade-off. Also, we will expand the utilization of experimental data for system identification. We hope to utilize current collected dynamical data from this work to improve the accuracy and utility of biomolecular models and discover unknown mechanistic details in biomolecular networks.

## Methods

### Bode plot and settling time computation using a linearized state space model

The general form of a state-space model in continuous time is written as[10]:

$$\dot{x} = Ax + Bu$$

$$y = Cx + Du$$

In our system shown in Fig. 1 and Eqs. (1)–(3), the A matrix is called the dynamic matrix, which is the Jacobian matrix of the system evaluated at steady state. The B matrix is the control matrix, in our system, it is a $3 \times 1$ matrix computed by $d\dot{x}/\gamma$ since parameter $\gamma$ is the only input of interest in our analysis. The C matrix is the sensor matrix. Since we define that only species B in Eq. (3) is observable, $D = [0, 0, 1]$. Finally, we set the direct term D to 0. The settling time and peak were computed with the MATLAB_R2019a function *stepinfo* for a step response and with function *impulse* for an impulse response; the Bode plot and the response magnitude of a low frequency disturbance was computed with the MATLAB_R2019a function *bode*.

### Model simulations

Equations and the parameters of the models used in Figs. 1 and 2 are presented in the main test. The 3rd model that generated Fig. 4d is in the Supplementary Information. All models were simulated by solving corresponding ODEs using the MATLAB_R2019a function *ode15s* over a set of discrete time steps using estimated parameters based on biological relevance (see Supplementary Information).

### Plasmid construction and purification

All plasmids used in this study were created using Golden Gate[37] assembly, Gibson assembly, or 3G assembly[38], with NEB®Turbo Competent E. coli as the cloning strain. Plasmids were purified using a Qiagen QIAprep Spin Miniprep Kit (Qiagen 27104). Linear fragments were gel extracted and purified using MinElute Gel Extraction Kit (Qiagen 28606).

### Strains, growth media and in-cell part characterization experiments

All experiments were performed in E. coli strain JS006 MG1655 $\Delta araC \Delta LacIKan^s$)[30] with either constitutively expressed RhlR or constitutively expressed CinR and RhlR[28] integrated into the chromosome with pOSIP integration plasmid into *O*-site containing Kanamycin resistance gene[39]. Plasmids with different constructs were transformed into the modified JS006 competent cells. Depending on the antibiotic resistance, cells were plated on LB + Agar plates containing 100 µg/ml carbenicillin or/and chloramphenicol, and incubated overnight at 37 °C. At least three colonies of each experimental condition were inoculated into 200 µL of LB containing carbenicillin in a 2 mL 96-well block, and grown overnight at 37 °C at 1000 rpm in a benchtop shaker. Four microliters of the overnight culture were added to 196 µL of M9 supplemented M9 media (1X M9 minimal salts, 0.5 µg/mL thiamine hydrochloride, 1.0% glucose, 0.1% casamino acids, 0.2 mM magnesium sulfate, 0.1 mM calcium chloride) containing carbenicillin and/or chloramphenicol and grown for 4 h at 37 °C at 1000 rpm.

For single time-point measurements (Fig. 3b, d, e), the sub-culture was then diluted 20 times with M9 media containing its appropriate antibiotics and AHL inducers, then grown in a 96 well plate for 6 h before the measurement of fluorescence (504 nm excitation, 540 nm emission) and optical density (OD, 600 nm). The two AHL inducers used in this work are N-(3-Oxotetradecanoyl)-L-homoserine lactone (Cin, Sigma O9264) and 3-Hydroxy-C4-HSL, N-(3-Hydroxybutanoyl)-L-homoserine lactone (Rhl, Sigma 74359). Experiment in Fig. 3b was grown with chloramphenicol, containing 0 µM or 200 µM of Rhl inducer; Experiment in Fig. 3d was grown with carbenicillin and chloramphenicol, containing 0 µM or 200 µM of Rhl inducer; Experiment in Fig. 3e was grown with carbenicillin and chloramphenicol with four inducer conditions: 0 µM Cin and 0 µM Rhl, 10 µM Cin and 0 µM Rhl, 0 µM Cin and 200 µM Rhl, and 10 µM Cin and 200 µM Rhl.

For time-course characterization measurements (Fig. 3c, f), the sub-culture was then diluted 20X to 200 µL with M9 media containing its appropriate antibiotics and inducers then grown on a 96 well plate in a BioTek Synergy H1 plate reader. The experiment in Fig. 3c was grown in M9 media with carbenicillin and 10 µM Cin for 5 h. Experiment in Fig. 3f was grown in M9 media with carbenicillin, 10 µM Cin and 200 µM Rhl for 12 h. The plate reader incubates the runs at 37 °C with maximum linear shaking. It measures fluorescence (504 nm excitation, 540 nm emission) and optical density (OD 600 nm) every 10 min.

For RBS screening experiments that resulted in the appropriate constructs shown in Fig. 3d, e, the plasmids were assembled with an ARL (Anderson RBS library) using 3G assembly[38]. Colonies were picked and grown in 200 µL LB with carbenicillin and chloramphenicol overnight at 37 °C at 1000 rpm on a benchtop shaker. Four microliters of the overnight culture were added to 196 µL of supplemented M9 media to grow to the exponential phase before the experiment. For the RBS screen experiment on CinR expression (Fig. 3e), each culture resulted from a single colony was diluted 20X and grown in M9 media, induced with either Rhl only, or Cin and Rhl. The colonies that appeared to be "off" with Rhl and "on" with Cin and Rhl were selected. For the RBS screen experiment on LacI expression (Fig. 3f), each culture resulted from a single colony was diluted 20X and grown in M9 media, induced with Rhl and Cin. Along with the screened colonies, biological triplicates of positive control with LacI(M) was measured to

provide screening reference (Fig. 3f). The colonies with signals that are apparent but lower than the positive control references were selected. All selected colonies for RBS screening were sequenced, the resulting RBS sequences were cloned into the constructs for verification with at least three biologically independent replicates.

### Dynamical perturbation experiments

All perturbation experiments were performed in JS006 E. coli with genome integrated RhlR. A blank ColE1 high copy plasmid is transformed with the $P_{Rhl/LacO}$ controlled cassette on a p15A backbone to serve as the negative control. Four colonies of negative control and 20 or 24 colonies of each of the four testing constructs showed in Fig. 4a were picked. Cultures were grown in 200 μL LB with carbenicillin and chloramphenicol overnight at 37 °C at 1000 rpm on a benchtop shaker. Four microliters of the overnight culture were added to 196 μL of M9 media and grown for four fours to prepare the sub-culture. The experiments started with a 20X dilution of the sub-culture into M9 media with carbenicillin and chloramphenicol, induced with Cin and Rhl. The plate reader incubates the runs at 37 °C with maximum linear shaking. It measures fluorescence (504 nm excitation, 540 nm emission) and optical density (OD, 600 nm) every 10 min. The first stage of the run lasts 5.5–7 h. After disturbances were introduced, both intact and perturbed cultures were grown in the plate reader for 2–3 h, and the fluorescent and OD dynamics were collected. Then, the disturbance of both groups were removed. Subsequently, the run continued in the plate reader for another 16–18 h. The detailed protocols of each perturbation are listed in Supplementary Table 7.

### Statistics & reproducibility

Data analysis was performed with Microsoft Excel and MATLAB_R2019a function *t* test. No statistical method was used to predetermine sample size. All single colonies were picked randomly from agar plates where the cells where transformed with purified circular DNA using antibiotics as selection markers. No manual group allocation methods were used. Each plate resulted from a single transformation and all colonies were assumed to be biological replicates. No data were excluded from the analyses in experiments presented in Fig. 3. Data exclusion was utilized in experiments presented in Figs. 4–6 only based on growth profiles, as described in Results and Supplementary Fig. 5.

### Reporting summary

Further information on research design is available in the Nature Research Reporting Summary linked to this article.

## Data availability

Source data for all main text and supplementary figures are available in the Source Data and Source Code folder at https://doi.org/10.6084/m9.figshare.20525034. Source data are provided with this paper.

## Code availability

Source code for all main text and supplementary figures are available in the Source Data and Source Code folder at https://doi.org/10.6084/m9.figshare.20525034.

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

## Acknowledgements

The authors would like to thank John Doyle, Fangzhou Xiao, Ayush Pandey, and Xinying Ren for their insightful discussions, as well as John Doyle, John Marken, and Ayush Pandey for their feedback on the manuscript. The author C.Y.H. is partially supported by Defense Advanced Research Projects Agency (Agreement HR0011-17-2-0008 - R.M.M.). The content of the information does not necessarily reflect the position or the policy of the government, and no official endorsement should be inferred.

## Author contributions

C.Y.H. and R.M.M. conceived and designed the study; C.Y.H. performed the control analysis and simulations, developed the code, performed all experiments, and wrote the manuscript. R.M.M. secured the funding and provided guidance.

## Competing interests

The authors declare no competing interests.
