## [Peer Review File · Nature Communications]

Reviewers' Comments:

Reviewer #1:

Remarks to the Author:

The manuscript presents simulation and experimental evidence which suggests that "layered" feedback control can overcome certain limitations on the performance control systems. While the manuscript is generally well written, there are several important conceptual points which are unclear and need to be addressed (detailed below). There are also minor grammatical errors throughout the manuscript that should be addressed.

1) Explanation of the efficiency-robustness tradeoff: The robustness efficiency tradeoff limit is not clearly explained in the manuscript. Specifically, in references [18] and [19], efficiency is not defined as the "If a system responses [sic] to changes in inputs quickly." In addition, the authors talk about both "inputs" and "disturbances." It is not clear if the authors are making a distinction between these or not, and whether they mean certain exogenous signals are inputs and others are disturbances, or if they are referring to e.g. different frequency ranges of the same exogenous signal. This conceptual point needs to be fixed, since it is central to the main claims of the manuscript. More generally, if the authors wish to show that their architecture exceeds a hard performance trade-off, then they need to be precise in defining the trade-off, especially since it is not clear the cited works back up the authors description of the design trade-off

2) Figure 1:

- i) It is not clear what the takeaway from the Bode plots is supposed to be. The bode plots are for a particular value of the parameters, and therefore it is not obvious that any results would generalize. In addition, while in the text the authors describe the plots, they do not say how they relate to either the figure title, "Stability analysis of the layered feedback controller using a minimal model," nor do they explain the connection to the efficiency robustness tradeoff.
- ii) From the authors' code it appears that the authors are plotting the settling time of the linearized system in the lower panels of 1D, but this is not mentioned in the text.
- iii) In Figure 1A it is not clear that either [20] or [21] justify the linear relationship between robustness and efficiency that the authors plot.

3) Robustness-efficiency Trade-off Analysis with Generic Biomolecular Configurations: It is not clear why in this section the authors chose to simulate systems with parameters randomly chosen from within 25% of the "nominal" values. A design tradeoff would typically be where no matter how the designer tunes the parameters some limit in the 2D performance space cannot be exceeded. Therefore the authors need to consider parameter values that are chosen purposefully to obtain high performance, not just random values near a single "guess."

4) While not strictly necessary, it would greatly strength the manuscript if the authors could exploit the mechanistic models the introduce to explain why a) some implementations do better than others and b) why layered control is exceeded the performance of the non layered systems.

5) In Figure 5 the authors plot T40 which they define. However, the "if a trajectory failed to recover 40%, we took the final time measurement as its T40" seems like an odd choice. Not only would the nature choice be $T_{40} = \text{infinity}$ if the system never recovers by 40%, but since the final time is different between the different experiments it is hard for the reader to tell from looking at the plots which systems failed to recover.

6) In Figure 5, it is not clear the layered control is the best in panels B, C, E, and F. This contradicts the authors' statement that "Finally, in eight sets of different dynamical perturbation experiments, we observed that the layered control feedback consistently overcomes the robustness-efficiency trade-off limit."

Reviewer #2:

Remarks to the Author:

In this paper the authors explore the possible implementation of layered feedback loops in

biomolecular regulatory circuits as a means to overcome possible limits in efficiency-robustness tradeoffs. They use modelling of basic two node regulatory motifs covering open, cis, trans, and layered regulation and explore the response dynamics of these systems to input perturbations when biologically realistic parameters and interaction forms are used. They show that layered feedback is able to overcome efficiency-robustness tradeoffs and go on to build genetic implementations of these systems. These show that layered feedback in some cases can improve response dynamics.

Overall, this is a very interesting paper that makes excellent use of a set of simple regulatory motifs to demonstrate some key control-related results in biology. I very much enjoyed reading it and can see that it will resonate widely with the synthetic biology community. The paper is on the whole well-written (although the grammar in places should be checked), the mathematical analysis is sound, but the experimental work is less convincing given the limitations of the approaches they have used, and very large variability seen in the response dynamics. That said, the results are very interesting and the application of control theory to engineered biology is of great interest at present to the community. I would recommend that the authors consider my major comments and think about how the experimental aspects could be further strengthened or theoretical elements expanded to make the work more comprehensive and supportive of the papers overall goal. At present, the idea is nice and is backed up by theory, but it let down by the experiments presented.

Major comments:

1. For the initial analysis of the model it would be useful to provide a derivation of the linearized system and equilibrium points in the SI. Also, I really struggled to understand what precise parameters were used for to generate the plots in Figure 1D and none of the axes are labelled. I understand this analysis was more interested in the qualitative features of the systems and that code is provided, but even so, the simulation output and data presented must be able to be quantitatively understood from the main text.

2. My biggest concern relates to the experimental results and methodology. The use of cell washing and resuspension into fresh media is incredibly difficult to do reproducibly as varying amounts of inducer will remain in the cells and they will also likely respond to the environmental shock of being taken out of 37C. It is likely that these difficulties are born out in the very large variability that is seen across replicates in Figure 5 that to my mind make it difficult to rigorously compare the difficult regulatory approaches. While there are definitely patterns in the responses measured, it is often the case that layered feedback does not perform best for the majority of replicates. The design of the constructs looks sound, but I'd suggest trying to use some form of continual culture system to remove the need to shock the cells or consider carrying out far more replicates to better gauge the actual variability that is due to this part in the experiment. The difficult I see at present is that the result the theory suggest is only really seen if cherry picking the results from the experiments and that more evidence is needed to be sure what the "average" behavior of these constructs is.

3. Page 15: "This work provided experimental proof..." This work does not provide proof experimentally at all. It presents results that suggest layered feedback *may* be beneficial and the results show that in some limited cases that to be true, but the experiments are too preliminary to make such strong statements. Please consider rewording the Discussion/Conclusion to present the work in a more factually correct light.

Other minor comments:

4. Page 2: "If a system responses to change" -> "If a system responds to change"

5. Page 3: "for faster degradation" -> "to capture faster degradation"

6. Page 3: "we constraint the product" -> "we constrain the product"

7. Page 5: "we analyze the disturbance" -> "we analyzed the disturbance". Please check the voice throughout as it varies within sections.

8. Page 8: Could references of where the estimates have been taken from be included where possible. Needs some evidence that they are biologically realistic.
9. Page 11: "four testing constructs" -> "four test constructs"
10. Page 11: "as a bacterial culture grows towards capacity". What does this mean, please clarify and use appropriate terminology? Check this throughout as you've used similar terminology later on as well.
11. Page 11: what evidence for protein degradation rate increasing when nearing stationary phase. There are many assumptions made in this paragraph that are not backed up by any experimental citations.
12. Page 14: "an 8 unit moving average"? Do you mean 8 time point moving average.
13. Methods: "using guessed or estimated parameters"? Not sure the term guessed is suitable for a method. Are they not estimated based on the knowledge you have as well?
14. Methods: Please include full genotype of strains when mentioned in Methods.
15. Methods: Please include full details of the software used (including versions) and all the parameters chosen during simulation in the Methods. At present it would not be possible to reproduce the results precisely. I realize the code is provided, but these are details that should be present in the main text.

Reviewer #1 (Remarks to the Author):

The manuscript presents simulation and experimental evidence which suggests that "layered" feedback control can overcome certain limitations on the performance control systems. While the manuscript is generally well written, there are several important conceptual points which are unclear and need to be addressed (detailed below). There are also minor grammatical errors throughout the manuscript that should be addressed.

1) Explanation of the efficiency-robustness tradeoff: The robustness efficiency tradeoff limit is not clearly explained in the manuscript. Specifically, in references [18] and [19], efficiency is not defined as the "If a system responses [sic] to changes in inputs quickly." In addition, the authors talk about both "inputs" and "disturbances." It is not clear if the authors are making a distinction between these or not, and whether they mean certain exogenous signals are inputs and others are disturbances, or if they are referring to e.g. different frequency ranges of the same exogenous signal. This conceptual point needs to be fixed, since it is central to the main claims of the manuscript. More generally, if the authors wish to show that their architecture exceeds a hard performance trade-off, then they need to be precise in defining the trade-off, especially since it is not clear the cited works back up the authors description of the design trade-off

We thank the reviewer for pointing out the confusion on the definition of the trade-off. Throughout the revised manuscript, we have changed the trade-off from "robustness-efficiency" to "robustness-speed" to reflect its definition. In addition, we now define the trade-off between speed and robustness in the Introduction section.

2) Figure 1:

We appreciate the reviewer's critiques and suggestions on this figure; the responses are included below:

i) It is not clear what the takeaway from the Bode plots is supposed to be. The bode plots are for a particular value of the parameters, and therefore it is not obvious that any results would generalize. In addition, while in the text the authors describe the plots, they do not say how they relate to either the figure title, "Stability analysis of the layered feedback controller using a minimal model," nor do they explain the connection to the efficiency robustness tradeoff.

The Bode plot serves two purposes:

1. The output magnitude of the two proposed architectures (R as an RNA regulator vs. R as a protein regulator) when responding to low-frequency disturbances are indistinguishable. The two designs of R are simulated with consistent and effective repression strength. ($K_R=10$, $dR=0.3$ for RNA, $K_R=100$, $dR=0.03$ for protein). In the revised manuscript, we have added text for clarification in the Discussion section.
2. A graphic demonstration of what the top panel in 1D is plotting. We have also added a sentence in the Figure 1 caption to clarify this.

We have modified the Figure caption title and added text to the discussion to connect the figures to the concept of performance trade-off.

ii) From the authors' code it appears that the authors are plotting the settling time of the linearized system in the lower panels of 1D, but this is not mentioned in the text.

We have added text in the discussion to clarify the linearization. In addition, we added a section in Method to walk through this procedure.

iii) In Figure 1A it is not clear that either [20] or [21] justify the linear relationship between robustness and efficiency that the authors plot.

This plot is only meant to illustrate the trade-off concept for a broad audience with various backgrounds. We do not intend to claim a linear relationship between the two metrics. We have now clarified this in the Figure 1 caption.

3) Robustness-efficiency Trade-off Analysis with Generic Biomolecular Configurations: It is not clear why in this section the authors chose to simulate systems with parameters randomly chosen from within 25% of the "nominal" values. A design tradeoff would typically be where no matter how the designer tunes the parameters some limit in the 2D performance space cannot be exceeded. Therefore, the authors need to consider parameter values that are chosen purposefully to obtain high performance, not just random values near a single "guess."

We thank the reviewer for bringing up this confusion. Parameter tuning is nontrivial in biology. We are often stuck with a set of parameters with little flexibility. Therefore, instead of choosing parameters to optimize the system's performance, we are interested in how a certain system design behaves with a given set of parameters. The

parameters used in this work were based on previous parameterization work and relevant biological parameters from the literature. We chose a 25% range to generate parameters to introduce a certain level of randomness without breaking the biological relevance. In the revised manuscript, we have added clarification on the parameter choices in the Discussion section. In addition, we added a section in the Supplemental Information with more literature-supported details to walk through the biological relevance of key parameters.

4) While not strictly necessary, it would greatly strength the manuscript if the authors could exploit the mechanistic models the introduce to explain why a) some implementations do better than others and b) why layered control is exceeded the performance of the non layered systems.

We appreciate the reviewer's suggestion. We agree the experimental results raised many unanswered questions. For example, it is unclear why the trans, cis and the layered feedback controls often respond to the same perturbation in different directions; it is also unclear why the trans feedback outperforms the cis feedback under one type of perturbation and vice versa with another kind of perturbation. We are also interested in explaining how gene expression dynamics link to growth dynamics. All of these could be explored with mechanistic models. However, this is not trivial work since we don't have all the details in gene expression dynamics in living bacterial cells. We are working on a separate project to pursue the answers to the questions mentioned above.

5) In Figure 5 the authors plot T40 which they define. However, the "if a trajectory failed to recover 40%, we took the final time measurement as its T40" seems like an odd choice. Not only would the nature choice be $T_{40} = \infty$ if the system never recovers by 40%, but since the final time is different between the different experiments it is hard for the reader to tell from looking at the plots which systems failed to recover.

We thank the reviewer for bringing up this issue. We agree that this metric we used was unclear. Therefore, we repeated the experiments in the original Figure 5 with more time points and more replicates. The speed metric has also been updated to be consistent with control textbooks to be the "settling time" that describes the time it takes for a trajectory to recover back to the defined error band after the start of a perturbation. If a trajectory fails to recover back into the error band, we used high order polynomial fit to predict the time when it recovers. We also used heatmaps to illustrate the systems dynamics of all individual replicates of four constructs under each type of perturbation. Figures 4-6 in the revised manuscript are now updated to showcase the new data.

6) In Figure 5, it is not clear the layered control is the best in panels B, C, E, and F. This contradicts the authors' statement that "Finally, in eight sets of different dynamical perturbation experiments, we observed that the layered control feedback consistently overcomes the robustness-efficiency trade-off limit."

We thank the reviewer for pointing out this inconsistency. We have updated all disturbance experiments to include more replicates, so it is possible to compute the p-values to support the statistical significance. The updated version now consists of 6 sets of disturbance experiments (Figure 5 and 6, Figure S4~S8). Two of them are chemical (AHL-Cin and AHL-Rhl inducers) perturbations, two of them are temperature perturbations, and two are nutrient (glucose concentration) perturbations. We decided to remove the Rhl perturbations to simplify the data analysis and discussion, as its representation of chemical perturbation is redundant. We have also updated the claims throughout the manuscript to reflect the statistically supported conclusions.

Reviewer #2 (Remarks to the Author):

In this paper the authors explore the possible implementation of layered feedback loops in biomolecular regulatory circuits as a means to overcome possible limits in efficiency-robustness tradeoffs. They use modelling of basic two node regulatory motifs covering open, cis, trans, and layered regulation and explore the response dynamics of these systems to input perturbations when biologically realistic parameters and interaction forms are used. They show that layered feedback is able to overcome efficiency-robustness tradeoffs and go on to build genetic implementations of these systems. These show that layered feedback in some cases can improve response dynamics.

Overall, this is a very interesting paper that makes excellent use of a set of simple regulatory motifs to demonstrate some key control-related results in biology. I very much enjoyed reading it and can see that it will resonate widely with the synthetic biology community. The paper is on the whole well-written (although the grammar in places should be checked), the mathematical analysis is sound, but the experimental work is less convincing given the limitations of the approaches they have used, and very large variability seen in the response dynamics. That said, the results are very interesting and the application of control theory to engineered biology is of great interest at present to the community. I would recommend that the authors consider my major comments and think about how the experimental aspects could be further strengthened or theoretical elements expanded to make the work more comprehensive and supportive of the papers overall goal. At present, the idea is nice and is backed up by theory, but it let down by the experiments presented.

Major comments:

1. For the initial analysis of the model it would be useful to provide a derivation of the linearized system and equilibrium points in the SI. Also, I really struggled to understand what precise parameters were used for to generate the plots in Figure 1D and none of the axes are labelled. I understand this analysis was more interested in the qualitative features of the systems and that code is provided, but even so, the simulation output and data presented must be able to be quantitatively understood from the main text.

We thank the reviewer for bringing up this confusion. We have updated the Figure 1 caption and added a subsection in Methods to clarify the procedures we took to obtain the linearized state-space model and the MATLAB functions to compute the system's Bode plot and settling time. Additionally, the parameters for the 3-equation model are now included in the main text.

2. My biggest concern relates to the experimental results and methodology. The use of cell washing and resuspension into fresh media is incredibly difficult to do reproducibly as varying amounts of inducer will remain in the cells and they will also likely respond to the environmental shock of being taken out of 37C. It is likely that these difficulties are born out in the very large variability that is seen across replicates in Figure 5 that to my mind make it difficult to rigorously compare the difficult regulatory approaches. While there are definitely patterns in the responses measured, it is often the case that layered feedback does not perform best for the majority of replicates. The design of the constructs looks sound, but I'd suggest trying to use some form of continual culture system to remove the need to shock the cells or consider carrying out far more replicates to better gauge the actual variability that is due to this part in the experiment. The difficult I see at present is that the result the theory suggest is only really seen if cherry picking the results from the experiments and that more evidence is needed to be sure what the "average" behavior of these constructs is.

We thank the reviewer for bringing up this issue. We agree that we need more replicates to achieve a statistically significant conclusion. The disturbance experiments are now all updated. In the revised manuscript, we included six sets of disturbance experiments with more replicates (20 replicates for open loop, 24 replicates for each feedback-controlled case). Two of these experiments are chemical perturbations (Cin+Rhl dip and spike), two are environmental perturbations (temperature dip and spike), and two are nutrient perturbations (glucose dip and spike). We decided to remove the Rhl disturbance to simplify the data analysis and discussion, as its representation of chemical perturbations is redundant. We performed two-tail t-tests for the robustness and speed metrics and marked whether the differences were statistically significant. We have also added heatmaps to aid the visualization of the dynamical profile of all replicates (Figure 5A and Figure S4~S8). Figure 5 now shows the system disturbance with AHL wash. The data of the five other sets of experiments are now in Figure 6 and Figure S4~S8.

3. Page 15: "This work provided experimental proof..." This work does not provide proof experimentally at all. It presents results that suggest layered feedback *may* be beneficial and the results show that in some limited cases that to be true, but the experiments are too preliminary to make such strong statements. Please consider rewording the Discussion/Conclusion to present the work in a more factually correct light.

We thank the reviewer for pointing this out. We have updated the Abstract, Introduction, Discussion, and Result sections with factual claims that are supported by the experiments.

Other minor comments:

We thank the reviewer for the minor comments, the responses are included below:

4. Page 2: "If a system responses to change" -> "If a system responds to change"

The manuscript has been updated to reflect this correction.

5. Page 3: "for faster degradation" -> "to capture faster degradation"

The manuscript has been updated to reflect this correction.

6. Page 3: "we constraint the product" -> "we constrain the product"

The manuscript has been updated to reflect this correction.

7. Page 5: "we analyze the disturbance" -> "we analyzed the disturbance". Please **check the voice** throughout as it varies within sections.

The manuscript has been updated to reflect this correction.

8. Page 8: Could references of where the estimates have been taken from be included where possible. Needs some evidence that they are biologically realistic.

We have added a section in the Supplemental Information to walk through the parameter choices and their biological relevance.

9. Page 11: “four testing constructs” -> “four test constructs”

The manuscript has been updated to reflect this correction.

10. Page 11: “as a bacterial culture grows towards capacity”. What does this mean, please clarify and use appropriate terminology? Check this throughout as you’ve used similar terminology later on as well.

We have updated “capacity” to “stationary phase” throughout the manuscript.

11. Page 11: what evidence for protein degradation rate increasing when nearing stationary phase. There are many assumptions made in this paragraph that are not backed up by any experimental citations.

We have included citations in the section to back up these claims.

12. Page 14: “an 8 unit moving average”? Do you mean 8 time point moving average.

Yes, we have updated this section.

13. Methods: “using guessed or estimated parameters”? Not sure the term guessed is suitable for a method. Are they not estimated based on the knowledge you have as well?

Yes, the guesses are based on biological relevance. We have updated the text to reflect that.

14. Methods: Please include full genotype of strains when mentioned in Methods.

The genotype of the strain JS006 has been added into the Methods section.

15. Methods: Please include full details of the software used (including versions) and all the parameters chosen during simulation in the Methods. At present it would not be possible to reproduce the results precisely. I realize the code is provided, but these are details that should be present in the main text.

There were three models involved in this manuscript. The equations and the parameters of the 1st and the 2nd models are now presented in the main manuscript file. The 3rd model that generated Figure 4D is in the Supplemental Information. We have updated the Methods section to clarify this information.

Reviewers' Comments:

Reviewer #1:

Remarks to the Author:

Layered Feedback Control Overcomes Performance Trade-off in Synthetic Biomolecular Networks by Chelsea Y. Hu and Richard M. Murray

General comments to authors (round 2):

The revised manuscript has clarified some points of confusion. However, several conceptual issues have remained unaddressed, and with some of the revisions new conceptual questions have been raised. First, as detailed below, the authors have not fixed the conceptual issue with the claimed design tradeoff between speed and robustness. In essence, the manuscript does not clearly define the tradeoff that is being investigated, and additionally the references that the authors cite when introducing the design tradeoff do not support the existence of a "speed-robustness" tradeoff that can be overcome with layered control. Second, while it is still unclear how the authors are defining robustness, there is a definite mismatch between what they investigate in the first part of the paper and the later parts. In figure one the authors use the steady state response to a constant disturbance as a measure of robustness, whereas in the later parts they use the maximum value of the response to a spike disturbance. These are not the same, and so any evidence provided in Figure 1 that there might in fact be a tradeoff that layered control can overcome is not relevant to the experimental results. In addition to these problems, the following conceptual point is very important: The authors central claim is that there is a tradeoff between robustness and speed which can be overcome with layered feedback. This would mean, as the authors illustrate in Figure 1A, that there is a curve in the speed-robustness plane such that all control strategies that are not layered must lie above the curve. The authors then provide three points in this plane, two without layering and one with layering. There are significant conceptual issues with using this evidence to prove the authors' claim:

- i) Clearly, two non-layered points are not able to show a curve that all non layered systems must lie above, since no matter the values of the two points one could simply draw a line between them.
- ii) Simply showing that the layered system is both faster and more robust than either of the two non-layered systems cannot prove that the layered system overcomes the tradeoff, since the two non-layered systems might not be "on" the tradeoff curve, instead both might be strictly above the curve, and the layered one exactly on it.

In order to support their claim experimentally, the authors would need to provide multiple non-layered systems with at the bare minimum different strengths of cis and trans feedback, and show that with cis and trans feedback together the speed and robustness are better than **any** of the systems with only cis or trans feedback. This could be done by parameter tuning in the experiments.

In addition to the conceptual points above, the reviewer still has doubts about some of the experimental results because the variation between replicates is very high.

Please see the specific follow ups to each of the reviewer's original comments, detailed below:

Specific comments:

Reviewer #1 (Remarks to the Author):

The manuscript presents simulation and experimental evidence which suggests that "layered" feedback control can overcome certain limitations on the performance control systems. While the manuscript is generally well written, there are several important conceptual points which are unclear and need to be addressed (detailed below). There are also minor grammatical errors throughout the manuscript that should be addressed.

1) Explanation of the efficiency-robustness tradeoff: The robustness efficiency tradeoff limit is not clearly explained in the manuscript. Specifically, in references [18] and [19], efficiency is not defined as the "If a system responses [sic] to changes in inputs quickly." In addition, the authors talk about both "inputs" and "disturbances." It is not clear if the authors are making a distinction between these or not, and whether they mean certain exogenous signals are inputs and others are disturbances, or if they are referring to e.g. different frequency ranges of the same exogenous signal. This conceptual point needs to be fixed, since it is central to the main claims of the manuscript. More generally, if the authors wish to show that their architecture exceeds a hard performance trade-off, then they need to be precise in defining the trade-off, especially since it is not clear the cited works back up the authors description of the design trade-off

We thank the reviewer for pointing out the confusion on the definition of the trade-off. Throughout the revised manuscript, we have changed the trade-off from “robustness-efficiency” to “robustness-speed” to reflect its definition. In addition, we now define the trade-off between speed and robustness in the Introduction section.

Response:

The authors have not satisfactorily addressed this comment. The authors' claim is still poorly defined and not supported by the references. To define a design tradeoff in control systems, one must first precisely define the system being considered in terms of inputs, outputs, and the structure of the connections between the subsystems. Then, one must define what is meant by robustness and speed in terms of the properties of specific signals in the system. The authors have not done this, and the cited references do not support the authors' claim. Reference [18] does not discuss a tradeoff between speed and robustness. Reference [19] does not consider layered vs nonlayered feedback. Reference [20] discusses neuroscience, and does not make any claims about trade-offs and layered control in general. Reference [21] does not discuss “layers” in the sense that this manuscript considers them. [21] does not discuss speed-accuracy (or similar) tradeoffs in control systems.

2) Figure 1:

We appreciate the reviewer's critiques and suggestions on this figure; the responses are included below:

i) It is not clear what the takeaway from the Bode plots is supposed to be. The bode plots are for a particular value of the parameters, and therefore it is not obvious that any results would generalize. In addition, while in the text the authors describe the plots, they do not say how they relate to either the figure title, "Stability analysis of the layered feedback controller using a minimal model," nor do they explain the connection to the efficiency robustness tradeoff.

The Bode plot serves two purposes:

1. The output magnitude of the two proposed architectures (R as an RNA regulator vs. R as a protein regulator) when responding to low-frequency disturbances are indistinguishable. The two designs of R are simulated with consistent and effective repression strength. ($K_R=10$, $d_R=0.3$ for RNA, $K_R=100$, $d_R=0.03$ for protein). In the revised manuscript, we have added text for clarification in the Discussion section.
2. A graphic demonstration of what the top panel in 1D is plotting. We have also added a sentence in the Figure 1 caption to clarify this.

We have modified the Figure caption title and added text to the discussion to connect the figures to the concept of performance trade-off.

Response:

Resolved.

ii) From the authors' code it appears that the authors are plotting the settling time of the linearized system in the lower panels of 1D, but this is not mentioned in the text.

We have added text in the discussion to clarify the linearization. In addition, we added a section in Method to walk through this procedure.

Response:

Resolved.

iii) In Figure 1A it is not clear that either [20] or [21] justify the linear relationship between robustness and efficiency that the authors plot.

This plot is only meant to illustrate the trade-off concept for a broad audience with various backgrounds. We do not intend to claim a linear relationship between the two metrics. We have now clarified this in the Figure 1 caption.

Response:

Figure 1A is not supported by the references or the presented results. This conceptual issue is related to the reviewer's major comment 1) above. Essentially, the authors have not properly defined what they mean by a tradeoff

between “robustness” and “speed,” and have additionally not presented valid evidence in the form of references or otherwise that there exists such a general tradeoff, and that it can be overcome by layering.

3) Robustness-efficiency Trade-off Analysis with Generic Biomolecular Configurations: It is not clear why in this section the authors chose to simulate systems with parameters randomly chosen from within 25% of the "nominal" values. A design tradeoff would typically be where no matter how the designer tunes the parameters some limit in the 2D performance space cannot be exceeded. Therefore, the authors need to consider parameter values that are chosen purposefully to obtain high performance, not just random values near a single "guess."

We thank the reviewer for bringing up this confusion. Parameter tuning is nontrivial in biology. We are often stuck with a set of parameters with little flexibility. Therefore, instead of choosing parameters to optimize the system's performance, we are interested in how a certain system design behaves with a given set of parameters. The parameters used in this work were based on previous parameterization work and relevant biological parameters from the literature. We chose a 25% range to generate parameters to introduce a certain level of randomness without breaking the biological relevance. In the revised manuscript, we have added clarification on the parameter choices in the Discussion section. In addition, we added a section in the Supplemental Information with more literature-supported details to walk through the biological relevance of key parameters.

Response:

The authors have misinterpreted this comment. The authors claim that there is a control theoretic trade-off between robustness and speed. Such a trade-off would mean that a non-layered controller cannot exceed some combination of robustness and speed no matter how it is tuned, not that a non-layered controller for a specific biological system would not exceed some combination of robustness and speed.

4) While not strictly necessary, it would greatly strengthen the manuscript if the authors could exploit the mechanistic models they introduce to explain why a) some implementations do better than others and b) why layered control exceeds the performance of the non-layered systems.

We appreciate the reviewer's suggestion. We agree the experimental results raised many unanswered questions. For example, it is unclear why the trans, cis and the layered feedback controls often respond to the same perturbation in different directions; it is also unclear why the trans feedback outperforms the cis feedback under one type of perturbation and vice versa with another kind of perturbation. We are also interested in explaining how gene expression dynamics link to growth dynamics. All of these could be explored with mechanistic models. However, this is not trivial work since we don't have all the details in gene expression dynamics in living bacterial cells. We are working on a separate project to pursue the answers to the questions mentioned above.

Response:

Resolved

5) In Figure 5 the authors plot T40 which they define. However, the "if a trajectory failed to recover 40%, we took the final time measurement as its T40" seems like an odd choice. Not only would the nature choice be $T_{40} = \infty$ if the system never recovers by 40%, but since the final time is different between the different experiments it is hard for the reader to tell from looking at the plots which systems failed to recover.

We thank the reviewer for bringing up this issue. We agree that this metric we used was unclear. Therefore, we repeated the experiments in the original Figure 5 with more time points and more replicates. The speed metric has also been updated to be consistent with control textbooks to be the “settling time” that describes the time it takes for a trajectory to recover back to the defined error band after the start of a perturbation. If a trajectory fails to recover back into the error band, we used high order polynomial fit to predict the time when it recovers. We also used heatmaps to illustrate the systems dynamics of all individual replicates of four constructs under each type of perturbation. Figures 4-6 in the revised manuscript are now updated to showcase the new data.

Response:

The revised version of Figure 5 does not support the authors' claims. The large variation in settling time between the different replicates (especially for the layered and trans cases) indicates that that experimental results are not repeatable. Additionally, the authors provide no evidence that fitting a polynomial curve when the system fails to recover within the specified time window is a valid analysis method. Such a technique needs to be justified, since

setting aside issues with the accuracy of the settling time estimates generated in this way, the system may very well fail to every recover, which would undermine the experimental results.

6) In Figure 5, it is not clear the layered control is the best in panels B, C, E, and F. This contradicts the authors' statement that "Finally, in eight sets of different dynamical perturbation experiments, we observed that the layered control feedback consistently overcomes the robustness-efficiency trade-off limit."

We thank the reviewer for pointing out this inconsistency. We have updated all disturbance experiments to include more replicates, so it is possible to compute the p-values to support the statistical significance. The updated version now consists of 6 sets of disturbance experiments (Figure 5 and 6, Figure S4~S8). Two of them are chemical (AHL-Cin and AHL-Rhl inducers) perturbations, two of them are temperature perturbations, and two are nutrient (glucose concentration) perturbations. We decided to remove the Rhl perturbations to simplify the data analysis and discussion, as its representation of chemical perturbation is redundant. We have also updated the claims throughout the manuscript to reflect the statistically supported conclusions.

Response:

The data in panel A does not support the authors' claims. First, the "speed" data is inconsistent between replicates and seems to indicate a lack of repeatability. Second, the layered control does not recover faster than the open loop system, which indicates there is a serious, unexplained problem with the experiment. Additionally, looking at the trajectories in A, for the layered control sometimes the trajectory goes above zero after the disturbance and sometimes it does below zero, indicating a lack of reproducibility and probably a problem with the experimental design.

Reviewer #2:

Remarks to the Author:

I commend the authors on carefully considering and addressing my comments.

Layered Feedback Control Overcomes Performance Trade-off in Synthetic Biomolecular Networks by Chelsea Y. Hu and Richard M. Murray

General comments to authors (round 2):

The revised manuscript has clarified some points of confusion. However, several conceptual issues have remained unaddressed, and with some of the revisions new conceptual questions have been raised. First, as detailed below, the authors have not fixed the conceptual issue with the claimed design tradeoff between speed and robustness. In essence, the manuscript does not clearly define the tradeoff that is being investigated, and additionally the references that the authors cite when introducing the design tradeoff do not support the existence of a “speed-robustness” tradeoff that can be overcome with layered control. Second, while it is still unclear how the authors are defining robustness, there is a definite mismatch between what they investigate in the first part of the paper and the later parts. In figure one the authors use the steady state response to a constant disturbance as a measure of robustness, whereas in the later parts they use the maximum value of the response to a spike disturbance. These are not the same, and so any evidence provided in Figure 1 that there might in fact be a tradeoff that layered control can overcome is not relevant to the experimental results. In addition to these problems, the following conceptual point is very important: The authors central claim is that there is a tradeoff between robustness and speed which can be overcome with layered feedback. This would mean, as the authors illustrate in Figure 1A, that there is a curve in the speed-robustness plane such that all control strategies that are not layered must lie above the curve. The authors then provide three points in this plane, two without layering and one with layering. There are significant conceptual issues with using this evidence to prove the authors’ claim:

- i. Clearly, two non-layered points are not able to show a curve that all non layered systems must lie above, since no matter the values of the two points one could simply draw a line between them.
- ii. Simply showing that the layered system is both faster and more robust than either of the two nonlayered systems cannot prove that the layered system overcomes the tradeoff, since the two nonlayered systems might not be “on” the tradeoff curve, instead both might be strictly above the curve, and the layered one exactly on it.

In order to support their claim experimentally, the authors would need to provide multiple non-layered systems with at the bare minimum different strengths of cis and trans feedback, and show that with cis and trans feedback together the speed and robustness are better than **any** of the systems with only cis or trans feedback. This could be done by parameter tuning in the experiments.

In addition to the conceptual points above, the reviewer still has doubts about some of the experimental results because the variation between replicates is very high.

We appreciate the reviewer's comments and suggestions for improving this work. In the 2nd round of revision, we made the following major changes:

1. We improved and corrected the writing in the Introduction section and revised the concept illustration to clarify definitions of the speed and robustness trade-off. The original Figure 1A was not meant to illustrate “that there is a curve in the speed-robustness plane such that all control strategies that are not layered must lie above the curve.” As we showed in the original Figure 1D, the robustness and speed of single layers with strong repression and slow degradation can outperform the layered version with weak repression and fast degradation (comparing the southwest corners of singles and the northeast corners of the layered design). The line was meant to illustrate that when the performance of two given single controllers falls on this line, then layering them together could overcome the limit bound by this line. We thank the reviewer for pointing out this conceptual confusion to us, we now have updated the illustration of trade-off to reflect our definition of this concept (new Figure 1B), and we defined it in the legend that: with a given set of parameters that define regulator R, if one type of feedback is fast and fragile and another type of feedback is robust and slow, then layering these two feedbacks together could overcome the robustness-speed trade-off limit bound by these two feedbacks alone.

2. We revised Figure 1 to show system analysis with both step response and impulse response. We thank the reviewer for pointing out the inconsistency in step response and impulse response. In this revision, we chose to include both step response and impulse response. Because although the later simulation and experiments are subjected to impulse response, they are in low frequency. Therefore, we included both analyses to provide more insights using the linearized space state model.

3. We performed parameter tuning with the generic biomolecular model, now shown in Figure 2A. We appreciate the reviewer's suggestion on parameter tuning in the experiment to build multiple non-layered systems. Although doing so would certainly strengthen the conclusion of the manuscript, this is a nontrivial task, given the current status of synthetic biology. Engineering regulators with a variety of binding affinities and degradation rates requires an extensive amount of work on protein engineering and sRNA regulator engineering, which is out of the scope of this work. Instead, we performed this analysis in silico and included it in Figure 2A. We hope this simulated result can provide sufficient clarity to the audience.

4. We rewrote the discussion sections to clarify the results obtained in Figure 6, with added model simulation to help explain the results. The perturbation experiment shown in Figure 5 is meant to validate the model-predicted system performance by applying isolated perturbation in transcription. In Figure 6, we performed five more experiments to interrogate the synthetic biological system with complex perturbations. These complex perturbations are poorly defined yet highly relevant to biology. The reviewer's comments allowed us to realize the need to improve the manuscript structure and depth of the discussion. In this revision, we re-organized the sections and rewrote the discussion for Figure 6. Additionally, we added model simulation in Figure S6, S7, and S8 to aid the discussion of the results.

Please see the specific follow ups to each of the reviewer's original comments, detailed below:

Specific comments:

Reviewer #1 (Remarks to the Author):

The manuscript presents simulation and experimental evidence which suggests that "layered" feedback control can overcome certain limitations on the performance control systems. While the manuscript is generally well written, there are several important conceptual points which are unclear and need to be addressed (detailed below). There are also minor grammatical errors throughout the manuscript that should be addressed.

1. Explanation of the efficiency-robustness tradeoff: The robustness efficiency tradeoff limit is not clearly explained in the manuscript. Specifically, in references [18] and [19], efficiency is not defined as the "If a system responses [sic] to changes in inputs quickly." In addition, the authors talk about both "inputs" and "disturbances." It is not clear if the authors are making a distinction between these or not, and whether they mean certain exogenous signals are inputs and others are disturbances, or if they are referring to e.g. different frequency ranges of the same exogenous signal. This conceptual point needs to be fixed, since it is central to the main claims of the manuscript. More generally, if the authors wish to show that their architecture exceeds a hard performance trade-off, then they need to be precise in defining the trade-off, especially since it is not clear the cited works back up the authors description of the design trade-off

We thank the reviewer for pointing out the confusion on the definition of the trade-off. Throughout the revised manuscript, we have changed the trade-off from "robustness-efficiency" to "robustness-speed" to reflect its definition. In addition, we now define the trade-off between speed and robustness in the Introduction section.

Response:

The authors have not satisfactorily addressed to this comment. The authors' claim is still poorly defined and not supported by the references. To define a design tradeoff in control systems, one must first precisely define the system being considered in terms of inputs, outputs, and the structure of the connections between the subsystems. Then, one must define what is meant by robustness and speed in terms of the properties of specific signals in the system. The authors have not done this, and the cited references do not support the authors' claim. Reference [18] does not discuss a tradeoff between speed and robustness. Reference [19] does not consider layered vs nonlayered feedback. Reference [20] discusses neuroscience, and does not make any claims about trade-offs and layered control in general. Reference [21] does not discuss "layers" in the sense that this manuscript considers them. [21] does not discuss speed-accuracy (or similar) tradeoffs in control systems.

We thank the reviewer for pointing out the definition issue in this work. The system input and output are defined on page 4, line 6 in the tracked version: "Here we defined the output as species B. The input was defined as γ , a unitless scalar that impacts the expression rate of all three species (β_A , β_R ,) and β_B ." Additionally, we have also rewritten the text in the Introduction to define the robustness and speed: page

2, line 2: "Here we define \textit{speed} by the inverse of settling time and $\textit{robustness}$ by the inverse of peak disturbance of a system when subjected to an impulse or step perturbation."

We also appreciate the reviewer's attention to detail on the reference. We have rewritten the second paragraph in the Introduction section to reflect accurate descriptions of the references.

2. Figure 1:

We appreciate the reviewer's critiques and suggestions on this figure; the responses are included below:

i. It is not clear what the takeaway from the Bode plots is supposed to be. The bode plots are for a particular value of the parameters, and therefore it is not obvious that any results would generalize. In addition, while in the text the authors describe the plots, they do not say how they relate to either the figure title, "Stability analysis of the layered feedback controller using a minimal model," nor do they explain the connection to the efficiency robustness tradeoff.

The Bode plot serves two purposes:

1. The output magnitude of the two proposed architectures (R as an RNA regulator vs. R as a protein regulator) when responding to low-frequency disturbances are indistinguishable. The two designs of R are simulated with consistent and effective repression strength. ($KR=10$, $dR=0.3$ for RNA, $KR=100$, $dR=0.03$ for protein). In the revised manuscript, we have added text for clarification in the Discussion section.
2. A graphic demonstration of what the top panel in 1D is plotting. We have also added a sentence in the Figure 1 caption to clarify this.

We have modified the Figure caption title and added text to the discussion to connect the figures to the concept of performance trade-off.

Response:

Resolved.

ii. From the authors' code it appears that the authors are plotting the settling time of the linearized system in the lower panels of 1D, but this is not mentioned in the text.

We have added text in the discussion to clarify the linearization. In addition, we added a section in Method to walk through this procedure.

Response:

Resolved.

iii. In Figure 1A it is not clear that either [20] or [21] justify the linear relationship between robustness and efficiency that the authors plot.

This plot is only meant to illustrate the trade-off concept for a broad audience with various backgrounds. We do not intend to claim a linear relationship between the two metrics. We have now clarified this in the Figure 1 caption.

Response:

Figure 1A is not supported by the references or the presented results. This conceptual issue is related to the reviewer's major comment 1) above. Essentially, the authors have not properly defined what they mean by a tradeoff between "robustness" and "speed," and have additionally not presented valid evidence in the form of references or otherwise that there exists such a general tradeoff, and that it can be overcome by layering.

We thank the reviewer for pointing out the conceptual confusion caused by this illustration. We have now updated Figure 1 to include a more accurate illustration (Figure 1B). Additionally, we plotted out the simulated system performance in robustness and speed from the parameter scan presented by the heatmap (Figure E, F). We hope the new figure is sufficient to clarify the conceptual definition in this work.

3. Robustness-efficiency Trade-off Analysis with Generic Biomolecular Configurations: It is not clear why in this section the authors chose to simulate systems with parameters randomly chosen from within 25% of the "nominal" values. A design tradeoff would typically be where no matter how the designer tunes the parameters some limit in the 2D performance space cannot be exceeded. Therefore, the authors need to consider parameter values that are chosen purposefully to obtain high performance, not just random values near a single "guess."

We thank the reviewer for bringing up this confusion. Parameter tuning is nontrivial in biology. We are often stuck with a set of parameters with little flexibility. Therefore, instead of choosing parameters to optimize the system's performance, we are interested in how a certain system design behaves with a given set of parameters. The parameters used in this work were based on previous parameterization work and relevant biological parameters from the literature. We chose a 25% range to generate parameters to introduce a certain level of randomness without breaking the biological relevance. In the revised manuscript, we have added clarification on the parameter choices in the Discussion section. In addition, we added a section in the Supplemental Information with more literature supported details to walk through the biological relevance of key parameters.

Response:

The authors have misinterpreted this comment. The authors claim that there is a control theoretic trade-off between robustness and speed. Such a trade-off would mean that a non layered controller cannot exceed some combination of robustness and speed no matter how it is tuned, not that a non layered controller for a specific biological system would not exceed some combination of robustness and speed.

We appreciate the reviewer's explanation for clarifying this misinterpretation. We think this misunderstanding might be due to the original illustration in the old Figure 1A, and we hope the updated illustration can help clarify this problem. This work defines the trade-off by "specific" biological systems. There is no doubt that tuning parameters will achieve better performance, and one "good" single-layer feedback can outperform the layering of two "bad" feedback controls. This was demonstrated by the heatmap in Figure 1D in both the old and new versions and the new Figure S3: the southwest corner is always better than the northeast corner in both robustness (upper panel) and speed (lower panel). If we compare the most southwest grids in the single-layer control profiles, they are both faster and more robust than the northeast grids in the layered control map. We learned from the heatmaps that the performance of feedback control could be improved by making the regulator R both strongly repressive and slow-degrading. However, biomolecular species' binding affinity and degradation rates are highly inflexible and often non-ideal. In nature, cells often resort to layering multiple types of controllers together rather than evolving out a "perfect" regulator for single-layer controls. We hope our work will provide some insights into the design strategies of natural networks.

To further clarify the concept, we have updated Figure 2 on generic biomolecular configurations in this revision. In the new Figure 2A, we scanned regulator parameters for all four groups of designs. The resulting plots also showed the performance improvement of single-layer controls due to parameter tuning, and it is clear that layering further improves the system performance to a more optimal robustness-speed region in the trade-off plot.

While not strictly necessary, it would greatly strength the manuscript if the authors could exploit the mechanistic models they introduce to explain why a) some implementations do better than others and b) why layered control exceeds the performance of the non layered systems.

We appreciate the reviewer's suggestion. We agree the experimental results raised many unanswered questions. For example, it is unclear why the trans, cis and the layered feedback controls often respond to the same perturbation in different directions; it is also unclear why the trans feedback outperforms the cis feedback under one type of perturbation and vice versa with another kind of perturbation. We are also interested in explaining how gene expression dynamics link to growth dynamics. All of these could be explored with mechanistic models. However, this is not trivial work since we don't have all the details in gene expression dynamics in living bacterial cells. We are working on a separate project to pursue the answers to the questions mentioned above.

Response:

Resolved

4. In Figure 5 the authors plot T40 which they define. However, the "if a trajectory failed to recover 40%, we took the final time measurement as its T40" seems like an odd choice. Not only would the nature choice be $T_{40} = \infty$ if the system never recovers by 40%, but since the final time is different between the different experiments it is hard for the reader to tell from looking at the plots which systems failed to recover.

We thank the reviewer for bringing up this issue. We agree that this metric we used was unclear. Therefore, we repeated the experiments in the original Figure 5 with more time points and more replicates. The speed metric has also been updated to be consistent with control textbooks to be the "settling time" that describes the time it takes for a trajectory to recover back to the defined error band after the start of a perturbation. If a trajectory fails to recover back into the error band, we used high order polynomial fit to predict the time when it recovers. We also used heatmaps to illustrate the systems dynamics of all individual replicates of four constructs under each type of perturbation. Figures 4-6 in the revised manuscript are now updated to showcase the new data.

Response:

The revised version of Figure 5 does not support the authors' claims. The large variation in settling time between the different replicates (especially for the layered and trans cases) indicates that that experimental results are not repeatable. Additionally, the authors provide no evidence that fitting a polynomial curve when the system fails to recover within the specified time window is a valid analysis method. Such a technique needs to be justified, since setting aside issues with the accuracy of the settling time estimates generated in this way, the system may very well fail to every recover, which would undermine the experimental results.

We thank the reviewer's comments on the experimental results. The replicates shown in Figure 5 are biological replicates, meaning each trajectory is the sequential measurements of a liquid culture inoculated from a single bacterial colony. Biological replicates are inherently non-identical hence will introduce noise to the system. Also, gene expression in living cells is a combination of highly stochastic processes subjected to intrinsic and extrinsic noises within each replicate. We purposefully chose to plot all trajectories in heatmaps to highlight the biological uncertainties and the performance of feedback controls in this context. The settling time and robustness improvement have statistical significance as they are evaluated by the two-tail t-test (p-values marked in Figure 5D). Therefore, the variation between replicates here does not indicate lack of repeatability. On the contrary, it is an important piece of evidence that the experimental observation is repeatable – even in the stochastic context of biological systems.

Additionally, all trajectories in Figure 5 recovered within the experimental time window, no extrapolation method was used. Some systems do fail to recover, as shown in Figure 6, with complex perturbations. However, we think extrapolating settling time on these trajectories is still informative. Biological replicates are noisy, not every trajectory will behave as neatly as we hope. If we were to use no extrapolation method, then excluding "failed to recover" replicates would skew the result towards fast settling time. The polynomial curve fitting provides some quantification of settling time with all replicated trajectories included. If one replicate fails to recover by the end of the time window, the polynomial curve estimates a slower settling speed. In the context of binary qualitative comparison (i.e. faster or not), extrapolation is certainly fairer and more accurate than exclusion. We were also careful about over-extrapolation. For instance, in Figure 6E, almost all trajectories require extrapolation to quantify the settling time. It is unfair to compare settling time when no settling is observed in the time window. Hence, we did not compute this metric for that experiment.

5. In Figure 5, it is not clear the layered control is the best in panels B, C, E, and F. This contradicts the authors' statement that "Finally, in eight sets of different dynamical perturbation experiments, we observed that the layered control feedback consistently overcomes the robustness-efficiency trade-off limit."

We thank the reviewer for pointing out this inconsistency. We have updated all disturbance experiments to include more replicates, so it is possible to compute the p-values to support the statistical significance. The updated version now consists of 6 sets of disturbance experiments (Figure 5 and 6, Figure S4~S8). Two of them are chemical (AHL-Cin and AHL-Rhl inducers) perturbations, two of them are temperature perturbations, and two are nutrient (glucose concentration) perturbations. We decided to remove the Rhl perturbations to simplify the data analysis and discussion, as its representation of chemical perturbation is redundant. We have also updated the claims throughout the manuscript to reflect the statistically supported conclusions.

Response:

The data in panel A does not support the authors' claims. First, the "speed" data is inconsistent between replicates and seems to indicate a lack of repeatability. Second, the layered control does not recover faster than the open loop system, which indicates there is a serious, unexplained problem with the experiment. Additionally, looking at the trajectories in A, for the layered control sometimes the trajectory goes above zero after the disturbance and sometimes it does below zero, indicating a lack of reproducibility and probably a problem with the experimental design.

We thank the reviewer's comments on Figure 6 for helping us realize the need to improve the manuscript structure and depth of the discussion. Figure 6 is an "experimental exploration" because the dynamics in these disturbances are beyond what we can predict with mathematical models. In this revision, we renamed the subsection titles and regrouped Figure 4 and Figure 5 to be in the same section. We also rewrote the entire section to discuss Figure 6, with additional model simulations in Figure S6, S7, and S8 to aid the in-depth analysis of the results. For the results mentioned by the reviewer in Figure 6A, we showed that the layered feedback showed an improvement in robustness-speed, compared to the two single-layers (Figure S6D) with statistical significance ($p=0.014$ for layered to trans improvement, $p=0.0013$ for layer to cis improvement). Hence it supported our claim. We also point out that the open loop outperformed the single-layers, and we hypothesized that this is due to translational capacity capping, please see more details below.

To address the first point, all experimental replicates in this work were biological replicates inoculated from individual colonies. They are inherently non-identical due to "biological stochasticity." This is a very well-documented observation. The disturbance is a transcriptional "spike", in contrast to the transcriptional "dip" in Figure 5. Transcriptional "spike" is naturally noisier because its noise would also be amplified by translational bursting. Second, we respectfully disagree that unexpected observations indicate a "serious, unexplained problem with the experiment." In this observation, we hypothesize that the lack of improvement with feedback controls is caused by translational capacity capping. The open loop construct has a higher signal at the transcriptional level (more mRNA), which requires more translational machineries and energy to produce the system output protein. When the system is spiked with a higher concentration of inducers, the transcription is up-regulated (even more mRNA). However, since open loop already has the highest equilibrium readout, there might not be enough resources to manifest this spike at the transcriptional level; hence the effect of signal spike could be negated. In this revision, we rewrote the above discussion of this observation in-depth, along with additional model simulations to demonstrate our hypothesis (Figure S7). Additionally, although the observation in Figure 6A is noisy overall, the trajectories of all 21 replicates for layered control was relatively neat around 0, indicating highly robust system dynamics. More importantly, the conclusion we drew from these trajectories was statistically significant. Therefore, we respectfully disagree again that the result indicates a "lack of reproducibility and probably a problem with the experimental design." Biological networks are stochastic, complex, and often extremely messy. Yet it is the environment in which we study feedback controls. We designed these experiments to include and to highlight biological uncertainties, which we believe is the proper approach.

Reviewers' Comments:

Reviewer #1:

Remarks to the Author:

The reviewer thanks the authors for their careful edits and responses. The revised version of the manuscript is much improved, and the reviewer is mostly satisfied with the revised version. However, the following minor points need to be addressed:

1) A citation to [1] needs to be added. While it is a preprint on the arXiv, it discusses layering and the control theoretic tradeoff that the authors study here.

2) In the caption of Figure 1, the authors write: "With a given set of parameters that define R , if one type of feedback is fast and fragile and another type of feedback is robust and slow, then layering these two feedbacks together could overcome the robustness-speed trade-off limit bound by these two feedbacks alone." This sentence needs to be edited for clarity. In particular, it is unclear what precisely "the robustness-speed trade-off limit bound by these two feedbacks alone" refers to. Possibly it should be "...bound of either of these two feedbacks alone."

[1] Sarma, A. A., Li, J. S., Stenberg, J., Card, G., Heckscher, E. S., Kasthuri, N., Sejnowski, T & Doyle, J. C. (2021). Internal feedback in biological control: Architectures and examples. arXiv preprint arXiv:2110.05029.

The reviewer thanks the authors for their careful edits and responses. The revised version of the manuscript is much improved, and the reviewer is mostly satisfied with the revised version. However, the following minor points need to be addressed:

1) A citation to [1] needs to be added. While it is a preprint on the arXiv, it discusses layering and the control theoretic tradeoff that the authors study here.

We thank the reviewer for pointing us to this manuscript. We now have added it to our references as #33.

[1] Sarma, A. A., Li, J. S., Stenberg, J., Card, G., Heckscher, E. S., Kasthuri, N., Sejnowski, T & Doyle, J. C. (2021). Internal feedback in biological control: Architectures and examples. arXiv preprint arXiv:2110.05029.

2) In the caption of Figure 1, the authors write: "With a given set of parameters that define R, if one type of feedback is fast and fragile and another type of feedback is robust and slow, then layering these two feedbacks together could overcome the robustness-speed trade-off limit bound by these two feedbacks alone." This sentence needs to be edited for clarity. In particular, it is unclear what precisely "the robustness-speed trade-off limit bound by these two feedbacks alone" refers to. Possibly it should be "...bound of either of these two feedbacks alone."

We thank the reviewer for the help in improving this statement. We have updated this sentence in the caption to "With a given set of parameters that define R, if one type of feedback is fast and fragile and another type of feedback is robust and slow, then layering these two feedbacks together could overcome the robustness-speed trade-off limit bound by either of these two feedbacks alone."